# Temperature-Annealed Boltzmann Generators

**Henrik Schopmans** [1]   **Pascal Friederich** [1 2]

## Abstract

Efficient sampling of unnormalized probability densities such as the Boltzmann distribution of molecular systems is a longstanding challenge. Next to conventional approaches like molecular dynamics or Markov chain Monte Carlo, variational approaches, such as training normalizing flows with the reverse Kullback-Leibler divergence, have been introduced. However, such methods are prone to mode collapse and often do not learn to sample the full configurational space. Here, we present temperature-annealed Boltzmann generators (TA-BG) to address this challenge. First, we demonstrate that training a normalizing flow with the reverse Kullback-Leibler divergence at high temperatures is possible without mode collapse. Furthermore, we introduce a reweighting-based training objective to anneal the distribution to lower target temperatures. We apply this methodology to three molecular systems of increasing complexity and, compared to the baseline, achieve better results in almost all metrics while requiring up to three times fewer target energy evaluations. For the largest system, our approach is the only method that accurately resolves the metastable states of the system.

## 1. Introduction

Machine learning, and particularly generative models, have become a transformative force across numerous domains. A prime example of this impact is in structural biology, where deep learning methods such as those in the AlphaFold family (Jumper et al., 2021; Abramson et al., 2024) have revolutionized our ability to predict protein structures. While a big part of AlphaFold's success can surely be attributed to

an advanced methodology, a key factor also lies in the availability of abundant experimental data, such as that in the Protein Data Bank (PDB) (Burley et al., 2021).

However, not all scientific domains benefit from such well-curated and extensive experimental datasets. In areas where data scarcity is a persistent challenge, computational simulations play an essential role. Molecular dynamics (MD) and Markov chain Monte Carlo (MCMC) methods are the primary tools used to explore complex biochemical and physical systems and generate insights from limited experimental information. Despite their utility, these classical sampling approaches often come with significant computational costs, as they rely on iterative trajectory-based exploration of high-dimensional state spaces.

As a result, various approaches have been explored to speed up these methods, including integrating machine learning (ML)-based force fields (Reiser et al., 2022), enhanced sampling techniques (Barducci et al., 2011), and data-driven collective variables (Bonati et al., 2021). Furthermore, (transferable) generative models have been trained on equilibrium samples from MD simulations (Noé et al., 2019; Mahmoud et al., 2022; Zheng et al., 2024; Klein & Noe, 2024).

While these advancements have significantly improved the efficiency and utility of traditional simulations, there is a growing interest in rethinking the paradigm altogether. Variational sampling methods, rooted in generative modeling, offer a compelling alternative to classical MD and MCMC. These approaches aim to learn the underlying probability distribution without the availability of training data, bypassing the need for explicit trajectory-based sampling.

The most straightforward variational approach is to train a likelihood-based generative model, such as a normalizing flow, using the reverse Kullback-Leibler divergence (KLD). However, this is known to yield mode collapse in many scenarios (Midgley et al., 2023b; Felardos et al., 2023). Recently, multiple variational sampling methods were developed (Blessing et al., 2024), based on normalizing flows (Matthews et al., 2022; Midgley et al., 2023b), diffusion models (Zhang & Chen, 2021; Richter & Berner, 2023; Berner et al., 2023; Vargas et al., 2023; Zhang et al., 2023; Akhound-Sadegh et al., 2024; Sendera et al., 2024), and flow matching (Woo & Ahn, 2024).

[1]Institute of Theoretical Informatics, Karlsruhe Institute of Technology, Kaiserstr. 12, 76131 Karlsruhe, Germany [2]Institute of Nanotechnology, Karlsruhe Institute of Technology, Kaiserstr. 12, 76131 Karlsruhe, Germany. Correspondence to: Pascal Friederich <pascal.friederich@kit.edu>.

*Proceedings of the $42^{nd}$ International Conference on Machine Learning*, Vancouver, Canada. PMLR 267, 2025. Copyright 2025 by the author(s).

Despite their promise to accelerate sampling, the applicability and scalability of variational sampling methods remain limited, and the field is in its early stages of development compared to the wealth of research on hybrid MD/ML approaches. To the best of our knowledge, the only variational approach that has successfully been applied to the sampling of molecular systems with non-trivial multimodality, such as the popular benchmark system alanine dipeptide, is Flow Annealed Importance Sampling Bootstrap (FAB) (Midgley et al., 2023b).

In this work, we propose a novel and scalable flow-based framework to efficiently sample complex molecular systems without mode collapse. We train a normalizing flow at increased temperature using the reverse KLD, which we show reliably circumvents mode collapse. Since one is typically interested in the equilibrium distribution at lower temperatures, e.g. at room temperature, we introduce a reweighting-based training objective to iteratively anneal the distribution of the normalizing flow down to the target temperature. We demonstrate the capability and scalability of this methodology using three peptide systems of increasing complexity and achieve superior sampling efficiency and accuracy compared to baseline approaches.

Our contribution is threefold:

- We show that, in contrast to current literature, the reverse KLD is surprisingly powerful at learning the Boltzmann distribution of molecular systems without mode collapse, but only at increased temperatures where barriers between different free energy minima are lower and the probability distribution maxima are interconnected.

- We introduce an iterative reweighting-based training strategy to anneal the flow distribution to arbitrary target temperatures.

- We introduce two complex molecular systems as new benchmarks that go far beyond the size of the typically used benchmark system alanine dipeptide, and we demonstrate that our approach scales to those systems without mode collapse.

## 2. Related Work

Leveraging the improved mode-mixing behavior when sampling at higher temperatures is not a completely novel approach, as it was introduced as an accelerated sampling technique for MCMC and MD simulations before. Replica exchange Markov chain Monte Carlo (RE-MCMC) and molecular dynamics (REMD) use multiple parallel trajectories (replicas) at different temperatures, while allowing repeated exchanges of configurations between the replicas.

This essentially makes the high-temperature simulations help the lower-temperature simulations in overcoming slow energy barriers in the system.

Invernizzi et al. (2022) present a variation of replica exchange molecular dynamics using a normalizing flow that maps from the highest temperature directly to the target temperature. This allows direct exchanges and circumvents the need for intermediate replicas in the simulation. However, this still requires performing MD simulations at the boundary temperatures and it is not clear how well the method scales to larger systems.

Dibak et al. (2022) use a normalizing flow that is trained on samples from high-temperature molecular dynamics simulations. Using a special flow architecture, they show that the flow can be adapted to output low-temperature samples, even though it was only trained at the high temperature. Draxler et al. (2024) recently showed that the volume-preserving coupling layers used in that work are not universal, making the approach unsuitable for complex systems.

To solve this issue, Schebek et al. (2024) propose to use a normalizing flow with explicit conditioning on the temperature $T$ and pressure $P$. The prior of the normalizing flow is formed by samples from an MD simulation at a reference thermodynamic state $(T_0, P_0)$. The normalizing flow is subsequently trained to be able to sample across a range of thermodynamic states $(T, P)$ using the reverse KLD.

Similarly, Wahl et al. (2025) train a temperature-conditioned flow at increased temperature using samples from MD, and the correct temperature scaling to lower temperatures is obtained by matching the gradient of the unnormalized probability density of the flow with respect to the temperature to the gradient of the known target energy function.

So far, all mentioned approaches use (a large amount of) MD samples at at least one temperature, and transfer this to a different (lower) temperature. While a multitude of variational sampling methods that do not rely on samples from MD have been proposed, to the best of our knowledge, the only approach that has so far been successfully applied to non-trivial molecular systems is Flow Annealed Importance Sampling Bootstrap (FAB) by Midgley et al. (2023b).

Instead of the reverse KLD, FAB uses the $\alpha$-divergence with $\alpha = 2$ for energy-based training. The $\alpha$-divergence is estimated using annealed importance sampling (AIS) (Neal, 2001) from $q_X$ to $\frac{p_X^2}{q_X}$, where Hamiltonian Monte Carlo (HMC) (Duane et al., 1987) is used to transition between intermediate distributions. While AIS sampling is costly in terms of energy evaluations, it proved effective in learning complex probability distributions. They successfully learned the Boltzmann distribution of alanine dipeptide, a common benchmark molecular system, without mode collapse. However, how well this method scales to more com-

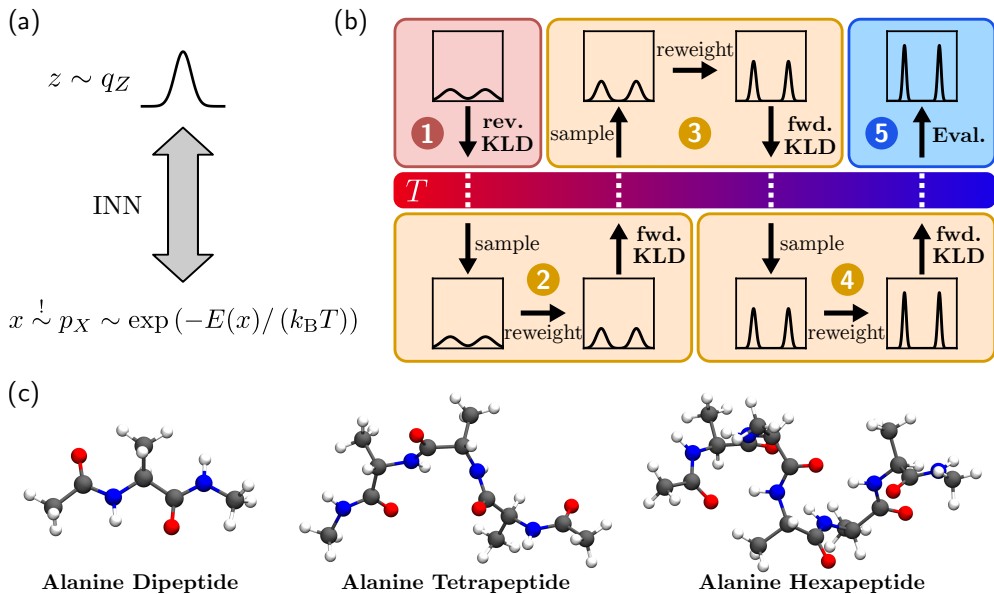

*Figure 1.* (a) Illustration of Boltzmann generators based on normalizing flows. The goal is to learn the equilibrium Boltzmann distribution of the 3D conformations of a molecular system. We focus on data-free training, where only the unnormalized probability density is known. (b) Illustration of our workflow. (1) To avoid mode collapse, we first train the flow at high temperature with the reverse KLD. (2-4) Then, the learned distribution is iteratively annealed to the target temperature by sampling at the current temperature, reweighting these samples to a lower temperature, and subsequently performing forward KLD training at this lower temperature. This is repeated multiple times until the desired target temperature is reached. (5) In the end, samples are drawn from the flow for evaluation. (c) Visualization of the three peptide molecular systems used in this work.

plex systems is unclear, and we will address this in this work when comparing our approach to FAB.

## 3. Preliminaries

### 3.1. Normalizing Flows

Normalizing flows use a latent distribution $q_Z(z)$, typically a Gaussian or uniform distribution, which is transformed to the target space using an invertible transformation $x = g(z; \theta)$, $z = f(x; \theta) = g^{-1}(x; \theta)$ (Figure 1a).

The transformed density of the flow can be expressed using the change of variables formula (Dinh et al., 2015):

$$q_X(x; \theta) = q_Z\left(f(x; \theta)\right) \left|\det J_{x \mapsto z}\right| \quad (1)$$

$$\text{with the Jacobian } J_{x \mapsto z} = \frac{\partial f(x; \theta)}{\partial x^T}$$

The most common approach to parameterize the invertible function is to use invertible coupling layers. In each coupling layer, the input $x_{1:D}$ is split into two parts $x_{1:d}$ and $x_{d+1:D}$. The first part is transformed elementwise conditioned on the second part, while the second part is kept

identical (see Figure 4 in the appendix for an illustration). If the elementwise transformation is invertible (monotonic), the whole transformation becomes invertible. Furthermore, the Jacobian matrix of such a coupling transform is lower triangular and can be efficiently computed (Durkan et al., 2019).

**Training with Samples.** The key property of normalizing flows is that the likelihood (Equation 1) is directly available, typically at the cost of a forward pass. This allows data-based maximum likelihood training (forward KLD):

$$\text{KL}_\theta\left[p_X \| q_X\right] = C - \int p_X(x) \log q_X(x; \theta)\, dx \quad (2)$$

$$= C - \mathbb{E}_{x \sim p_X}\left[\log q_Z\left(f(x; \theta)\right) + \log\left|\det J_{x \mapsto z}\right|\right] \quad (3)$$

**Training by Energy.** Next to data-based training, a normalizing flow can be trained by energy if only the target density $p_X(x)$ is known. In case of physical systems, such as the molecules studied in this work, this is the Boltzmann distribution $p_X(x) \sim \exp\left(-E(x)/(k_{\mathrm{B}}T)\right)$. Here, $x$ is the 3D configuration of the molecule, for example, the Cartesian coordinates of all atoms, $k_{\mathrm{B}}$ is the Boltzmann constant,

$T$ is the temperature, and $E(x)$ is the energy of the given configuration, evaluated either using quantum mechanics, e.g. with density functional theory, or, as in this work, using a parameterized force field. While other data-free training objectives exist (such as the $\alpha = 2$ divergence used in FAB), the reverse KLD is the most straightforward objective to fit the distribution of the flow to the target density, using samples from the flow itself:

$$\mathrm{KL}_\theta \left[ q_X \| p_X \right] = \mathrm{KL}_\theta \left[ q_Z \| p_Z \right] \qquad (4)$$

$$= C - \int q_Z(z) \log p_Z(z; \theta) \, \mathrm{d}z \qquad (5)$$

$$= C - \mathbb{E}_{z \sim q_Z} \left[ \log p_X \left( g(z; \theta) \right) + \log |\det J_{z \mapsto x}| \right] \quad (6)$$

**Importance Sampling** Since normalizing flows provide the likelihood of the generated samples, one can perform importance sampling to the true distribution $p_X$ using the importance weights $w(x) = \frac{p_X(x)}{q_X(x; \theta)}$. When estimating an expectation value of an observable $h(x)$ using samples $x_n$ from the flow distribution $q_X$, this offers asymptotically unbiased estimates (Martino et al., 2017; Noé et al., 2019):

$$\sum_{n=1}^{N} \frac{w(x_n)}{\sum_{i=1}^{N} w(x_n)} h(x_n) \xrightarrow[N \to \infty]{} \int h(x) p_X(x) \mathrm{d}x \quad (7)$$

While diffusion models and continuous normalizing flows can provide likelihoods by computing divergences of the involved vector field, this is often prohibitively expensive in practice, already for relatively small systems (Klein & Noe, 2024).

While Equation 7 theoretically allows unbiased estimates, this is limited in practice by the actual overlap between the flow distribution $q_X$ and the target distribution $p_X$. A helpful measure, here, is the effective sample size (ESS), defined as the number of independent samples needed from the target distribution $p_X$ to achieve the same variance in estimating expectation values as when using the flow distribution $q_X$ (Martino et al., 2017). The reverse ESS is an approximation of the ESS, where samples from the flow are used (see Section E in the appendix). The ESS is thus only estimated within the support of the flow, meaning that a high ESS can still be achieved if only parts of the true distribution are covered. Therefore, the reverse ESS value needs to always be interpreted together with other metrics.

## 4. Methods

### 4.1. Flow Architecture

Analogous to previous works (Midgley et al., 2023b; Schopmans & Friederich, 2024), we use an internal coordinate

representation based on bond lengths, angles, and dihedral angles to represent the molecular conformations. This incorporates the symmetries of the potential energy, which is invariant to translations and rotations of the whole molecule. For all experiments, we use a normalizing flow built from 16 monotonic rational-quadratic spline coupling layers (Durkan et al., 2019) with fully connected parameter networks in the couplings. Dihedral angles are treated using circular splines (Rezende et al., 2020) to incorporate the correct topology. Details can be found in Sections A and B of the appendix.

### 4.2. Temperature-Annealed Boltzmann Generators

Our approach to learn the Boltzmann distribution of molecular systems can be separated into two phases (see Figure 1b): First, we learn the distribution at a high temperature using the reverse KLD (step 1 in Figure 1b). Due to decreased barrier heights, this can be done without mode collapse. Secondly, the distribution is iteratively annealed using the mass-covering forward KLD with importance-sampled datasets to obtain the distribution at the target temperature. We now explain both steps in detail and discuss why they avoid the challenges described above.

#### Training by Energy: Avoiding Mode Collapse

The mode-seeking behavior of the reverse KLD has been discussed and observed in multiple previous publications (Midgley et al., 2023b; Felardos et al., 2023; Soletskyi et al., 2024). Once the flow collapsed to a mode, meaning that some remaining modes of the target distribution are not within the support of the flow, it will generally not escape this collapsed state if the remaining modes are too far separated from the collapsed mode. This is not surprising, since the reverse KLD is evaluated using an expectation value with samples from the flow distribution $q_X$ itself, which will only cover the collapsed modes.

At the typical target temperature for molecular systems, i.e. 300 K, modes are too far separated to be successfully covered with the reverse KLD. However, when sampling at increased temperature, the modes become more connected. Eventually, one can use the reverse KLD to efficiently learn the distribution. In this work, we performed the reverse KLD experiments at 1200 K, which allows a relatively small batch size and number of gradient descent steps to be used without mode collapse (see Section F.3 in the appendix for an extended ablation and discussion regarding choosing the starting temperature).

Using any loss function that directly includes the target energy of a molecular system can be challenging. If two atoms overlap sufficiently, the repulsive van der Waals energy diverges, leading to unstable training. Following previous work (Midgley et al., 2023b), we thus use a regularized energy function for training (see Section C in the appendix).

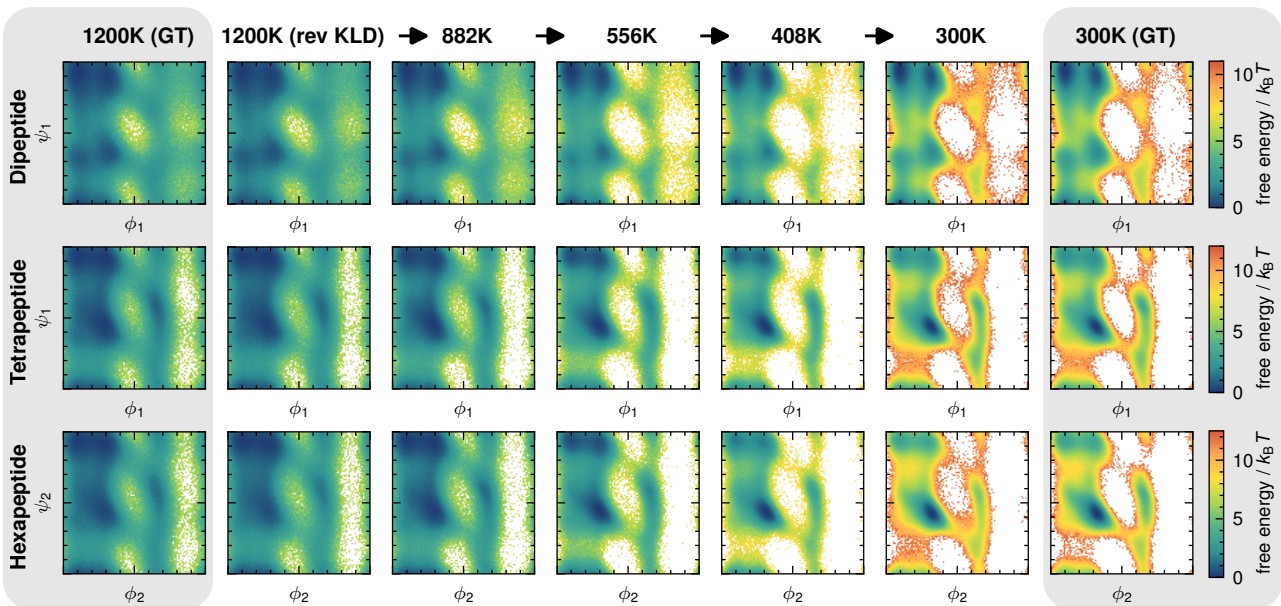

*Figure 2.* Visualization of the iterative annealing process, showing the free energy $F = -k_B T \ln p(\phi_i, \psi_i)$ of backbone dihedral angles (Ramachandran plots) in each iteration. After learning the distribution at 1200 K using the reverse KLD, the distribution is annealed step by step to the target temperature 300 K. Note that not all annealing iterations are shown. Since the tetrapeptide has three pairs of backbone dihedral angles, and the hexapeptide five, we selected one pair of dihedral angles for this illustration. The annealing of all pairs of backbone dihedrals can be found in Figures 13 and 16 in the appendix. We used $1 \times 10^7$ samples for the Ramachandran plots at 300 K and $1 \times 10^6$ samples for the rest.

This avoids very high values in the loss function and stabilizes training. Furthermore, analogous to previous work (Schopmans & Friederich, 2024), we found that removing a small fraction of the largest energy values in the loss contributions of each batch stabilizes training.

**Reweighting-Based Annealing**

As explained, we use the reverse KLD as a first step to learn the Boltzmann distribution at increased temperature. To obtain the distribution at a lower target temperature, here 300 K, we utilize importance sampling (Equation 7). While one could do importance sampling directly from $T_{\text{start}} = 1200$ K to $T_{\text{target}} = 300$ K, this will yield bad overlap and sampling efficiency (see Figure 5 in the appendix).

Instead, we perform importance sampling using multiple temperatures $T_1 = T_{\text{start}}, T_2, T_3, \ldots, T_{K-1}, T_K = T_{\text{target}}$, where $T_1 \geq T_2 \geq T_3 \geq \ldots \geq T_K$. In one annealing iteration, we perform the following steps:

1. Sample a dataset $\mathcal{D}_i = \{x_j\}_{j=1}^N$ of $N$ samples $x_j \sim q_X(x_j; \theta)$ from the flow at the current temperature $T_i$.

2. Calculate importance weights $w(x_j) = \frac{p_{X, T=T_{i+1}}(x_j)}{q_X(x_j; \theta)}$ for each sample $x_j$ in $\mathcal{D}_i$ to transition to $T_{i+1}$.

3. According to these importance weights, resample a dataset $\mathcal{W}_{i+1}$ with replacement from $\mathcal{D}_i$.

4. Perform forward KLD training on $\mathcal{W}_{i+1}$.

Throughout this annealing workflow, we keep updating the flow parameters, without reinitialization. In this way, we can anneal the distribution of the flow step by step toward the target temperature $T_{\text{target}}$ (see steps 2-4 in Figure 1b). Since we use the mass-covering forward KLD objective based on importance-sampled datasets, mode collapse is not a problem during the annealing.

For all experiments, we chose 9 temperature annealing iterations. To ensure a similar overlap between two consecutive distributions, we choose the temperatures $T_i$ using a geometric progression between $T_{\text{start}}$ and $T_{\text{target}}$ (Sugita & Okamoto, 1999) (see Section F.3 in the appendix for a comparison with a linear temperature schedule):

$$T_i = T_{\text{start}} \left( \frac{T_{\text{target}}}{T_{\text{start}}} \right)^{\frac{i-1}{K-1}} \quad (8)$$

Furthermore, we added a final fine-tuning iteration, where we sample at 300 K and reweight to 300 K ($T_{K-1} = T_K =$

$T_{\text{target}}$). Since the temperature is not lowered in this fine-tuning step, the effective sample size of the training dataset is higher, which empirically improves the final metrics obtained at 300 K. For the hexapeptide system, we added such a fine-tuning iteration with $T_{i+1} = T_i$ after each annealing iteration (see Equation 14 in the appendix). While this increases the total number of target potential energy evaluations, it improves the obtained results substantially. For the two less complex systems, intermediate fine-tuning was not necessary. We refer to Section F.3 of the appendix for an extended discussion and ablations regarding the fine-tuning iterations.

**Variations**

We note that our buffered reweighting approach is not the only option to anneal the temperature of a normalizing flow. As discussed in Section 2, a concurrent study (Wahl et al., 2025) uses a temperature-conditioned normalizing flow with a temperature scaling loss to learn the Boltzmann distribution at the target temperature. While this approach achieves promising results, it requires the repeated estimation of the partition function $Z$ using importance sampling with the flow distribution. Obtaining a low-variance estimate of $Z$ can be computationally expensive, especially for high-dimensional systems such as the hexapeptide studied here. However, a systematic comparison of different temperature scaling approaches is an interesting avenue for future work.

We further experimented with variations of our reweighting approach. For example, we tried training a temperature-conditioned flow with a reweighting-based objective continuously on the whole temperature range. This is described in more detail in Section 11 of the appendix. In practice, we found the iterative buffered annealing workflow to be superior, both in terms of accuracy and sampling efficiency.

## 5. Experiments

We now describe the conducted experiments to evaluate our temperature-annealing approach. The objective is to learn the Boltzmann distribution of three molecular systems, increasing in complexity (see Figure 1c). The first molecule is alanine dipeptide, a popular system that previously served as a benchmark for variational sampling (Midgley et al., 2023b), but also other related tasks such as transition path sampling (Holdijk et al., 2023; Seong et al., 2024). We further evaluate on two higher-dimensional and more complex systems, alanine tetrapeptide and alanine hexapeptide. All three systems have complex metastable high-energy regions that make up only a small fraction of the entire state space, which makes them suitable hard objectives for benchmarking.

While the focus of our work is on molecular systems due to

their challenging potential energy surface, TA-BG can also be applied to other sampling tasks. We refer to the appendix, Section K, for additional experiments on a Gaussian mixture system.

### 5.1. Baseline Methods

To judge the performance of our approach, we compare it to baseline methods. First, we trained a normalizing flow with the forward KLD using MD data from the target distribution. While this is not a variational sampling approach, it serves as a good baseline to show the expressiveness of the flow if data is available. Next, we trained a normalizing flow with the reverse KLD, targeting the Boltzmann distribution at 300 K. As the final and most powerful baseline, we trained Flow Annealed Importance Sampling Bootstrap (FAB) (Midgley et al., 2023b). As already discussed, to the best of our knowledge, this is the only method that so far has shown success without mode collapse on our smallest test system alanine dipeptide. It therefore serves as a strong baseline.

### 5.2. Metrics

To evaluate the distribution of the normalizing flow at the target temperature 300 K, we use a combination of multiple metrics. First, we use the negative log-likelihood (NLL) calculated on the ground truth test dataset. This is a good overall measure of the learned distribution. We note that since the metastable regions of our systems form only a small part of the ground truth test datasets, seemingly minor differences in the NLL can be decisive, especially when assessing mode collapse or the quality of the description of the metastable region.

To better assess potential mode collapse, we additionally evaluate the free energy $F = -k_{\text{B}}T \ln p(\phi_i, \psi_i)$ of the backbone dihedral angles (Ramachandran plots). Since these are the main slow degrees of freedom of the peptide systems, mode collapse will be directly visible here. To assess the quality of the Ramachandran plot, we calculate the forward KLD between the probability distribution given by the Ramachandran plot of the ground truth and the one of the flow distribution (RAM KLD). Since the tetrapeptide and hexapeptide have multiple pairs of backbone dihedral angles, we report the mean of their RAM KLD values. Furthermore, we repeat the calculation of the RAM KLD also using the Ramachandran plots obtained from importance sampling.

Last, to evaluate the sampling efficiency, we report the reverse effective sample size (ESS) (see Section 3).

## 6. Results

Figure 2 shows how the Ramachandran plots of each of the three systems are annealed to the target temperature,

showing four exemplary steps of the annealing workflow. Both the distribution at 1200 K learned with the reverse KLD and the distribution at the target temperature 300 K match the ground truth obtained from MD.

We now compare the obtained distribution at 300 K with that from the baseline methods by evaluating the introduced metrics (Table 1). Furthermore, Figure 3 shows the Ramachandran plots at 300 K obtained by all methods side-by-side. Figure 12 in the appendix shows the same comparison, but with importance sampling to the target distribution.

We start with the smallest system, alanine dipeptide. All methods, except for the reverse KLD, are able to learn the distribution without mode collapse (Figure 3). While the reverse KLD covers the high-energy region to some extent, partial mode collapse is visible. In terms of metrics, FAB and TA-BG obtain comparable results. Our method achieves better NLL and ESS values, while FAB achieves slightly lower RAM KLD values. Our approach only uses approximately a third of the target energy evaluations of FAB. We further note that the metrics we obtained with FAB on alanine dipeptide are slightly better but mostly comparable to those in the original publication.

Similar results can be observed for the tetrapeptide system. The reverse KLD training now collapses almost fully to the main mode, missing most of the metastable region (Figure 3). Both TA-BG and FAB achieve a good match with the ground truth distribution, fully resolving the metastable region. The metrics of TA-BG and FAB are again close, our approach achieves slightly better NLL and RAM KLD values, while FAB has a lower reweighted RAM KLD value and slightly higher ESS. Our approach again uses approximately a third of the target energy evaluations of FAB.

In the case of the most complex investigated system, alanine hexapeptide, the distribution of the reverse KLD training again almost entirely collapses. While FAB is partially able to resolve the metastable states, they differ in shape compared to the ground truth distribution. In contrast, our approach covers all metastable states without mode collapse and resolves them accurately with only small imperfections. This is also reflected and quantified by the metrics: Compared to FAB, our approach achieves better results in all metrics, while requiring $3.08 \times 10^8$ target energy evaluations compared to $4.2 \times 10^8$ used by FAB.

A tradeoff exists between the accuracy of the obtained distribution and the number of target evaluations. This is especially true for FAB, where the number of intermediate AIS distributions and the number of HMC steps can be varied. We present corresponding variations in Table 11 in the appendix. For FAB applied to the hexapeptide, even when using almost 3 times as many target evaluations compared to our approach, we still achieve a lower NLL value.

## 7. Discussion

To summarize, TA-BG achieves better results in most metrics compared to the baselines and thus learns the Boltzmann distribution of the investigated systems more accurately, while requiring significantly fewer target energy evaluations. We note that force field evaluations in our current setup are relatively inexpensive. With more accurate and computationally costly target evaluations, such as ML-based foundation models or density functional theory, the energy evaluation cost becomes dominant. In such cases, the higher sampling efficiency of our method will translate into significantly reduced computational cost. We refer to Section J in the appendix for a detailed analysis of the computational cost of TA-BG in comparison with FAB.

Next, we would like to point out the surprisingly good results we obtained by simply training with the reverse KLD at 300 K for alanine dipeptide (Figure 3). Even though at 300 K the metastable states are only connected to the global minimum by a very low-probability transition region, only partial mode collapse is observed. Next to our results of being able to successfully learn the distribution at high temperatures with the reverse KLD, this further shows the effectiveness of simple reverse KLD training.

Furthermore, previous work showed that not only training a normalizing flow from scratch with the reverse KLD is prone to mode collapse, but also fine-tuning a pre-trained flow to a target distribution with the reverse KLD typically collapses (Felardos et al., 2023). This is a not well understood phenomenon and has only recently been investigated theoretically (Soletskyi et al., 2024). While Felardos et al. (2023) proposed a solution to this problem, it is not clear how well it scales to larger systems. As described before, after our annealing workflow reaches 300 K, we fine-tune the flow distribution at 300 K by performing forward KLD training with a buffered dataset from the flow distribution, reweighted to the target distribution. This offers a simple yet effective solution to the problem of fine-tuning pre-trained flows and can also be used in other scenarios. For example, one can train a flow on biased MD simulation data that is not properly equilibrated, and then fine-tune with our buffered reweighting approach to obtain an unbiased flow distribution.

Furthermore, while our method generally yielded better results with fewer target energy evaluations for the hexapeptide, the results we obtained using FAB are still relatively close in terms of accuracy, especially for the other two systems. This establishes it as a powerful method for variational sampling. Therefore, a combination of our temperature-annealing approach with FAB is an interesting avenue for future work. Instead of using the reverse KLD for training at increased temperatures, FAB can be used and the distribution can be subsequently annealed with our TA-BG

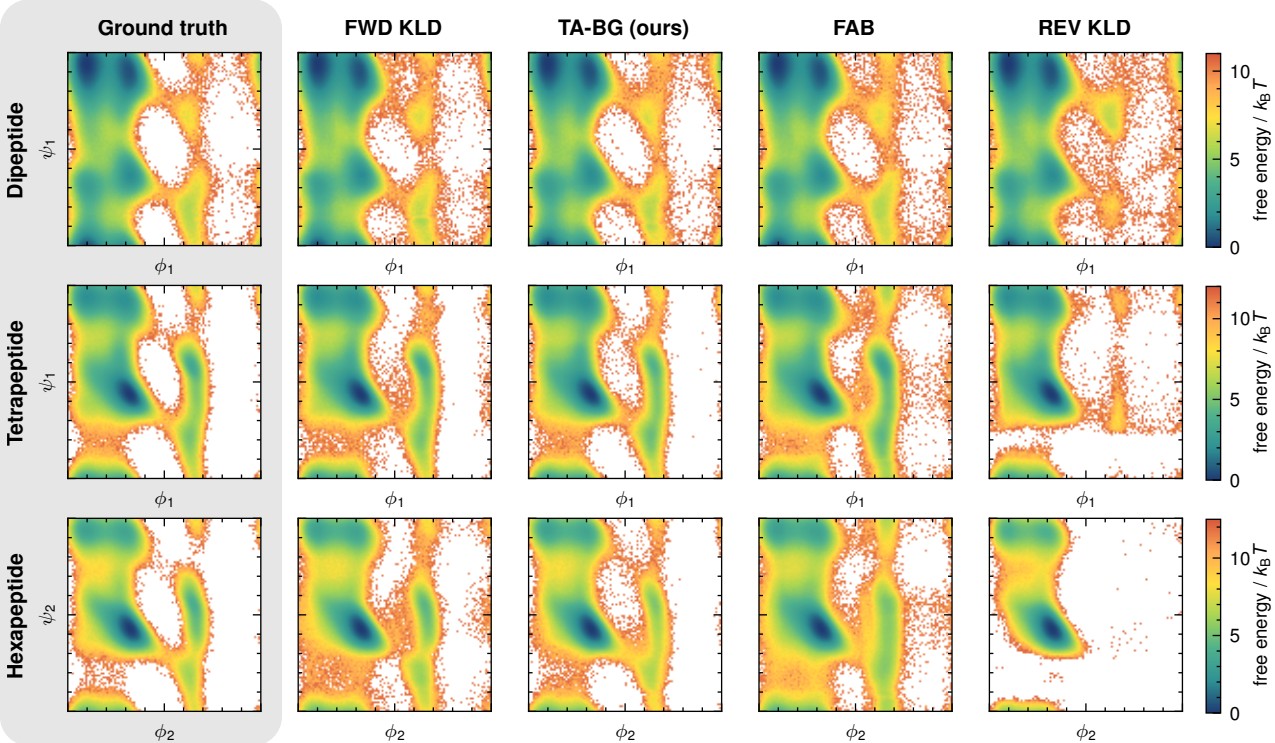

*Figure 3.* Comparison of the free energy $F = -k_\mathrm{B} T \ln p(\phi_i, \psi_i)$ of the backbone dihedral angles (Ramachandran plots) at 300 K obtained by the different methods. Since the tetrapeptide has three pairs of backbone dihedral angles, and the hexapeptide five, we selected the pair with the largest visible deviation among the methods. The Ramachandran plots for all pairs of backbone dihedral angles can be found in the appendix, Figures 14 and 17.

workflow. If and when this can be better than simple reverse KLD training at high temperatures needs to be investigated.

Our results indicate that other variational sampling approaches, such as those based on diffusion models, might benefit from sampling at higher temperatures and annealing the temperature afterwards. Other objectives will likely also benefit from the reduced mode separation at elevated temperatures.

In this work, we used a relatively simple flow architecture based on neural splines and internal coordinates. Recent improvements in flow architectures (Zhai et al., 2024; Tan et al., 2025) can be leveraged in the future to improve the obtained results and scale to larger systems. Furthermore, the use of an internal coordinate representation can be a limitation, since it is not transferable between systems and it does not incorporate the permutation invariance of identical particles. A transferable equivariant normalizing flow has recently been proposed (Midgley et al., 2023a). Using the TA-BG methodology in a transferable setting is an interesting avenue for future work, which may also allow a hybrid approach where data-free training is combined with data-based training (Lewis et al., 2024).

## 8. Conclusion

We introduced temperature-annealed Boltzmann generators, a technique that uses a combination of high-temperature reverse KLD pre-training and a subsequent annealing workflow to efficiently sample the Boltzmann distribution of molecular systems at room temperature. On the molecular systems investigated, our approach achieves better results in almost all metrics, while requiring up to three times fewer target energy evaluations compared to the baselines. Furthermore, it was the only variational approach that accurately resolved the metastable region of the most complex system studied, demonstrating its scaling capabilities from toy examples to application-relevant systems.

Similar to how replica exchange molecular dynamics is an established method in the toolbox of computational scientists, we are confident that high-temperature sampling with temperature-annealing is a powerful approach that will move the field of variational sampling forward.

*Table 1.* Comparison of metrics obtained for all three peptide systems. The table shows the number of potential energy evaluations (PE EVALS), the negative log-likelihood (NLL), the reverse effective sample size (ESS), the (mean) Ramachandran KLD (RAM KLD), and the (mean) Ramachandran KLD with reweighting (RAM KLD W. RW). All metrics are reported as the mean and standard error obtained from four independent experiments. The best-performing variational method for each metric is highlighted in bold.

| SYSTEM | METHOD | PE EVALS ↓ | NLL ↓ | ESS ↑ | RAM KLD ↓ | RAM KLD W. RW ↓ |
|---|---|---|---|---|---|---|
| DIPEPTIDE | FORWARD KLD | $5 \times 10^9$ | $-213.581 \pm 0.000$ | $(82.16 \pm 0.09)\,\%$ | $(2.21 \pm 0.05) \times 10^{-3}$ | $(1.99 \pm 0.07) \times 10^{-3}$ |
| | REVERSE KLD | $2.56 \times 10^8$ | $-213.609 \pm 0.006$ | $(94.11 \pm 0.21)\,\%$ | $(1.75 \pm 0.28) \times 10^{-2}$ | $(1.65 \pm 0.29) \times 10^{-2}$ |
| | FAB | $2.13 \times 10^8$ | $-213.653 \pm 0.000$ | $(94.81 \pm 0.04)\,\%$ | $\mathbf{(1.50 \pm 0.03) \times 10^{-3}}$ | $\mathbf{(1.25 \pm 0.01) \times 10^{-3}}$ |
| | TA-BG (OURS) | $\mathbf{7.56 \times 10^7}$ | $\mathbf{-213.665 \pm 0.002}$ | $\mathbf{(95.61 \pm 0.14)\,\%}$ | $(1.97 \pm 0.08) \times 10^{-3}$ | $(1.39 \pm 0.03) \times 10^{-3}$ |
| TETRA-PEPTIDE | FORWARD KLD | $4.2 \times 10^9$ | $-330.069 \pm 0.001$ | $(45.29 \pm 0.08)\,\%$ | $(2.26 \pm 0.06) \times 10^{-3}$ | $(2.50 \pm 0.03) \times 10^{-3}$ |
| | REVERSE KLD | $2.56 \times 10^8$ | $-329.191 \pm 0.122$ | $\mathbf{(74.88 \pm 3.65)\,\%}$ | $(3.00 \pm 0.35) \times 10^{-1}$ | $(2.87 \pm 0.40) \times 10^{-1}$ |
| | FAB | $2.13 \times 10^8$ | $-330.100 \pm 0.002$ | $(63.59 \pm 0.23)\,\%$ | $(6.89 \pm 0.25) \times 10^{-3}$ | $\mathbf{(1.25 \pm 0.01) \times 10^{-3}}$ |
| | TA-BG (OURS) | $\mathbf{7.56 \times 10^7}$ | $\mathbf{-330.114 \pm 0.004}$ | $(62.36 \pm 0.46)\,\%$ | $\mathbf{(2.61 \pm 0.20) \times 10^{-3}}$ | $(1.94 \pm 0.12) \times 10^{-3}$ |
| HEXA-PEPTIDE | FORWARD KLD | $4.2 \times 10^9$ | $-501.598 \pm 0.005$ | $(10.97 \pm 0.11)\,\%$ | $(4.16 \pm 0.26) \times 10^{-3}$ | $(7.69 \pm 0.03) \times 10^{-3}$ |
| | REVERSE KLD | $\mathbf{2.56 \times 10^8}$ | $-497.378 \pm 0.277$ | $\mathbf{(22.22 \pm 1.44)\,\%}$ | $(5.41 \pm 0.38) \times 10^{-1}$ | $(5.32 \pm 0.38) \times 10^{-1}$ |
| | FAB | $4.2 \times 10^8$ | $-501.268 \pm 0.008$ | $(14.64 \pm 0.08)\,\%$ | $(2.09 \pm 0.02) \times 10^{-2}$ | $(1.12 \pm 0.02) \times 10^{-2}$ |
| | TA-BG (OURS) | $3.08 \times 10^8$ | $\mathbf{-501.511 \pm 0.013}$ | $(14.71 \pm 0.18)\,\%$ | $\mathbf{(8.63 \pm 0.66) \times 10^{-3}}$ | $\mathbf{(8.77 \pm 0.47) \times 10^{-3}}$ |

## Software and Data

The source code to reproduce our experiments can be found at https://github.com/aimat-lab/TA-BG. Furthermore, the ground truth datasets from MD simulations are provided at https://doi.org/10.5281/zenodo.15526429.

## Acknowledgments

The authors would like to thank the anonymous reviewers for their valuable comments and suggestions. H.S. acknowledges financial support by the German Research Foundation (DFG) through the Research Training Group 2450 "Tailored Scale-Bridging Approaches to Computational Nanoscience". P.F. acknowledges funding from the Klaus Tschira Stiftung gGmbH (SIMPLAIX project) and the pilot program Core-Informatics of the Helmholtz Association (KiKIT project). Parts of this work were performed on the HoreKa supercomputer funded by the Ministry of Science, Research and the Arts Baden-Württemberg and by the Federal Ministry of Education and Research. This work is supported by the Helmholtz Association Initiative and Networking Fund on the HAICORE@KIT partition.

## Impact Statement

This paper presents work whose goal is to advance the field of Machine Learning. There are many potential societal consequences of our work, none which we feel must be specifically highlighted here.

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

## A. Internal Coordinate Representation

As discussed in the main text, we use an internal coordinate representation based on bond lengths, angles, and dihedral angles to represent the conformations of the molecular systems. As discussed in the next section, we use splines as the invertible transformations in our coupling blocks. These splines are only defined for mappings from the interval $[0, B]$ to $[0, B]$. Therefore, we scale all internal coordinates to fit in this range (here, $B = 1$).

To achieve this, we divide all dihedral angles by $2\pi$. Furthermore, bond lengths and angles are transformed as

$$\eta_i' = (\eta_i - \eta_{i;\min})/\sigma + 0.5 \,. \tag{9}$$

Here, $\eta_{\min}$ is the value of the corresponding degree of freedom from a minimum energy structure obtained from minimizing the initial structure with the force field. $\sigma$ was empirically chosen as $0.07\,\text{nm}$ for bond length dimensions and $0.5730\,\text{rad}$ for angle dimensions.

For the peptides studied in this work, two chiral forms (mirror images) exist. In nature, one almost exclusively finds only one of the two (L-form). However, since the potential energy is invariant to the mirror symmetry, there is no preference given by the energy model itself. Previous work (Midgley et al., 2023b; Schopmans & Friederich, 2024) simply filtered the "wrong" R-chirality during training. In contrast, we directly constrain generation to the L-chirality by restricting the output bounds of the splines that generate the dihedrals of the hydrogens at the chiral centers to the range $[0.5, 1.0]$. For this, we transform the corresponding dimensions as $\eta_i \to \eta_i \cdot 0.5 + 0.5$ after the flow generated them. This entirely removes the R-chirality from the space the flow can generate.

Furthermore, there is no preference given for the permutation of hydrogens in $CH_3$ groups. However, since the ground truth molecular dynamics simulations start from a given starting configuration, a preference does exist in the ground truth. Therefore, we restrict the generated distribution of the flow to this preference, by constraining the spline output range of the respective dihedral angles, analogous to how we constrain the chirality (see above).

## B. Architecture

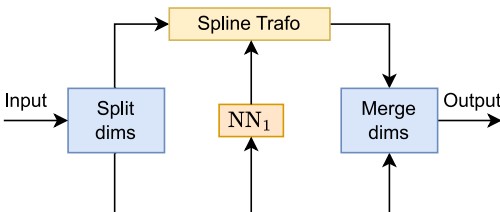

*Figure 4.* Illustration of a normalizing flow coupling layer.

For the normalizing flow architecture, we use an architecture similar to previous works (Midgley et al., 2023b; Schopmans & Friederich, 2024). As the invertible transformation in the coupling layers, we use monotonic rational-quadratic splines (Durkan et al., 2019) that map the interval $[0, 1]$ to $[0, 1]$ using monotonically increasing rational-quadratic functions with $K = 8$ bins.

We use 8 pairs of neural spline coupling layers. In each pair, we use a randomly generated mask to decide which dimensions to transform and which dimensions to condition the transformation on (see Figure 4). In the second coupling of each pair, the inverted mask is used. The dimensions of the dihedral angles are treated using circular splines (Rezende et al., 2020) to incorporate the correct topology. After each coupling layer, we add a random (but fixed) periodic shift to the dihedral angle dimensions.

The latent distribution $q_Z$ of the normalizing flow is a uniform distribution in the range $[0, 1]$ for the dihedral angle dimensions, and a Gaussian distribution with $\sigma = 0.5$ and $\sigma = 0.1$, truncated to the range $[0, 1]$, for the bond length and angle dimensions.

As the conditioning network ($NN_1$ in Figure 4), we use a fully connected neural network with hidden dimensions [256,256,256,256,256] and ReLU activation functions. Previous works used a fully connected neural network with a skip connection (Midgley et al., 2023b; Schopmans & Friederich, 2024), however, we found no benefit in this and therefore did not use a skip connection. To incorporate their periodicity, dihedral angles are represented as $(\cos \psi, \sin \psi)^\top$ before being passed as input to the neural network.

We used the same architecture for all experiments of all methods. The number of parameters in the architecture of each system can be found in Table 2. All experiments used the Adam optimizer (Kingma & Ba, 2017) to train the flow.

*Table 2.* Number of parameters in the normalizing flow architecture for each system.

|  | ALANINE DIPEPTIDE | ALANINE TETRAPEPTIDE | ALANINE HEXAPEPTIDE |
|---|---|---|---|
| NUMBER OF PARAMETERS | 7 421 512 | 9 452 376 | 12 124 616 |

To implement the normalizing flow models and the internal coordinate representations, we used the *bgflow* (Noé, 2024) and *nflows* (Conor Durkan et al., 2020) libraries with *PyTorch* (Paszke et al., 2019).

## C. Molecular Systems

*Table 3.* Overview of the molecular systems. The number of constrained bonds is given in brackets.

| NAME | SEQUENCE | NO. ATOMS | NO. HYDROGENS | NO. BONDS | NO. ANGLES | NO. TORSIONS |
|---|---|---|---|---|---|---|
| ALANINE DIPEPTIDE | ACE-ALA-NME | 22 | 12 | 21 | 20 | 19 |
| ALANINE TETRAPEPTIDE | ACE-3·ALA-NME | 42 | 22 | 19 (+ 22) | 40 | 39 |
| ALANINE HEXAPEPTIDE | ACE-5·ALA-NME | 62 | 32 | 29 (+ 32) | 60 | 59 |

To avoid diverging van der Waals energies due to atom clashes, we train with a regularized energy function (Midgley et al., 2023b):

$$E_{\text{reg.}}(E) =$$
$$\begin{cases} E, & \text{if } E \leq E_{\text{high}}, \\ \log(E - E_{\text{high}} + 1) + E_{\text{high}}, & \text{if } E_{\text{high}} < E \leq E_{\text{max}}, \\ \log(E_{\text{max}} - E_{\text{high}} + 1) + E_{\text{high}}, & \text{if } E > E_{\text{max}}. \end{cases} \tag{10}$$

For all systems, we used the energy regularization parameters $E_{\text{high}} = 1 \times 10^8$ and $E_{\text{max}} = 1 \times 10^{20}$.

Throughout this work, we use visualizations of the free energy $F = -k_{\text{B}}T \ln p(\phi_i, \psi_i)$ of the backbone dihedral angles of the systems (Ramachandran plots). For all Ramachandran plots, we used $1 \times 10^7$ samples if not otherwise specified. Furthermore, the axes are in scaled internal coordinates in the range $[0, 1]$.

## D. Force Field and Ground Truth Simulations

All ground truth simulations have been performed with OpenMM 8.0.0 (Eastman et al., 2024) using the CUDA platform. Ground truth energy evaluations during training have been performed with 18 workers in parallel using the OpenMM CPU Platform.

Details on the performed simulations and force field parameters for each system can be found in Table 4. For all systems, we used variants of Amber force fields (D.A. Case et al., 2023). For alanine dipeptide, the parameters are identical to the ones used in the FAB publication (Midgley et al., 2023b). We use the dataset made available by Stimper et al. (2022) as our test dataset. In addition to this ground truth test dataset, we performed another molecular dynamics simulation for alanine

dipeptide at 300 K (see Table 4). We used 50 ns for equilibration and a production simulation time of 5 µs. The small time step of 1 fs was chosen since this system does not use hydrogen bond length constraints. This additional simulation was used for training the forward KLD experiments and to create a separate validation dataset.

The force field parameters of the alanine tetrapeptide system match those used in the temperature steerable flow publication (Dibak et al., 2022). However, no public dataset for this system was available, which is why we performed two replica exchange molecular dynamics (REMD) simulations to obtain 300 K ground truth data. The hexapeptide system was, to the best of our knowledge, not used in previous publications, so we also here performed REMD simulations to obtain a ground truth dataset for evaluation. All REMD simulations used 200 ns equilibration without exchanges, 200 ns equilibration with exchanges, and 1 µs for the production simulation. For both the tetrapeptide and hexapeptide, we used one simulation for the ground truth test dataset and the other simulation to form the training dataset for the forward KLD experiments and to create a separate validation dataset.

Additionally to the simulations at 300 K, we performed high-temperature simulations at 1200 K to create validation datasets for the reverse KLD pre-training (see Table 4).

All datasets were subsampled randomly from the total production MD trajectories. The ground truth test datasets at 300 K contain $1 \times 10^7$ samples, the datasets used for the forward KLD experiments contain $1 \times 10^6$ samples. The additional validation datasets at 300 K and 1200 K contain $1 \times 10^6$ samples.

*Table 4.* Overview of the molecular dynamics simulations performed to obtain the ground truth datasets. We only specify the production simulation time. In the case of REMD simulations, this is the simulation time for each replica.

| SYSTEM | FORCE FIELD | CONSTRAINTS | $T$ / K | SIM. TIME / µs | TIME STEP / fs |
|---|---|---|---|---|---|
| ALANINE DIPEPTIDE | AMBER FF96 WITH OBC1 IMPLICIT SOLVATION | NONE | 300 1200 | 5.0 | 1.0 |
| ALANINE TETRAPEPTIDE | AMBER99SB-ILDN WITH AMBER99 OBC IMPLICIT SOLVATION | HYDROGEN BOND LENGTHS | 300, 332, 368, 408, 451, 500 (REMD) 1200 | 1.0 2.5 | 2.0 1.0 |
| ALANINE HEXAPEPTIDE | AMBER99SB-ILDN WITH AMBER99 OBC IMPLICIT SOLVATION | HYDROGEN BOND LENGTHS | 300, 332, 368, 408, 451, 500 (REMD) 1200 | 1.0 2.5 | 2.0 1.0 |

## E. Metrics Details

**RAM KLD**    To obtain comparable results to those in the original FAB publication (Midgley et al., 2023b), we evaluated the forward KLD of the Ramachandran plots in the same way. First, we calculated the probability density of the Ramachandran plot of the ground truth and that of the flow distribution on a 100x100 grid, using $1 \times 10^7$ samples for both. Then, the forward KLD is calculated between the two distributions.

**RAM KLD W. RW**    We repeat the same procedure to assess the obtained Ramachandran plot after reweighting to the target distribution. Here, analogous to Midgley et al. (2023b), we clipped the 0.01 % highest importance weights to the lowest value among them. This is necessary because of outliers in the importance weights due to flow numerics.

**ESS**    The ESS was calculated according to the following equation (Midgley et al., 2023a):

$$\frac{n_{\text{e,rv}}}{N} = \frac{1}{N \sum_{i=1}^{N} \bar{w}\left(x_i\right)^2} \tag{11}$$

$$\text{with} \quad x_i \sim q_X\left(x_i; \theta\right), \quad \bar{w}(x_i) = \frac{w(x_i)}{\sum_{i=1}^{N} w(x_i)}$$

Also here, we clipped the 0.01 % highest importance weights to the lowest value among them. Furthermore, the ESS was calculated with respect to the regularized energy function (Equation 10).

While one can also calculate the forward ESS, which uses samples from the ground truth (Midgley et al., 2023a), in practice we found this metric to yield spurious results, depending heavily on the chosen importance weight clipping value. Therefore, we chose to only use the reverse ESS, even though it does not capture mode collapse.

## F. TA-BG

### F.1. Workflow Variations

As described in the main text, next to our buffered iterative annealing workflow, we also tried training a temperature-conditioned normalizing flow on the whole continuous temperature range. To fix the distribution at 1200 K to the distribution learned by the reverse KLD, we used a split architecture to train with temperature-conditioning:

- A base flow generates samples at 1200 K. This was trained with the reverse KLD at 1200 K, and the model parameters of this base were frozen afterwards.

- A temperature-conditioned head flow is added to the base flow, which transforms the high-temperature samples to lower temperatures. The spline couplings of the head flow are scaled in such a way that they always output the identity for $T = 1200$ K.

In each batch, we sampled a support temperature $T_{\text{support}}$ and a target temperature $T_{\text{reweight}}$ and performed reweighted forward KLD training:

$$\mathcal{L}_{\text{reweight}} = - \mathop{\mathbb{E}}_{x \sim q_X(x;T_{\text{support}};\theta)} \frac{\exp\left(\frac{-E(x)}{k_{\text{B}} T_{\text{reweight}}}\right)}{\underbrace{q_X(x;T_{\text{support}};\theta)}_{\text{Stopped gradients}}} \log q_X(x;T_{\text{reweight}};\theta) \tag{12}$$

We also used this training objective with self-normalized importance weights within each batch, and with resampling each batch according to the importance weights.

The training objective in Equation 12 is similar to the one introduced by Wahl et al. (2025), but does not require the estimate of the partition function. In practice, we found the iterative annealing workflow to yield more accurate results, while also being more sampling efficient compared to using Equation 12.

A systematic comparison of using Equation 12 and using the training objective introduced by Wahl et al. (2025) can be explored in future work.

### F.2. Hyperparameters of Main Experiments

*Table 5.* Hyperparameters of the TA-BG experiments, annealing from 1200 K to 300 K. The cosine annealing learning rate scheduler is applied within each annealing iteration, so the learning rate resets in the beginning of each new annealing iteration. "Annealing iteration" here also refers to the fine-tuning iterations with $T_{i+1} = T_i$.

| | DIPEPTIDE | TETRAPEPTIDE | HEXAPEPTIDE |
|---|---|---|---|
| GRADIENT DESCENT STEPS PER ANNEALING ITERATION | 30 000 | 20 000 | 20 000 |
| LEARNING RATE | $5 \times 10^{-6}$ | $1 \times 10^{-5}$ | $5 \times 10^{-6}$ |
| BATCH SIZE | 2048 | 4096 | 2048 |
| LR SCHEDULER | COSINE ANNEALING | - | - |
| BUFFER SAMPLES DRAWN PER ANNEALING ITERATION | $5 \times 10^6$ | $5 \times 10^6$ | $1 \times 10^7$ |
| BUFFER RESAMPLED TO | $2 \times 10^6$ | $2 \times 10^6$ | $2 \times 10^6$ |

In all TA-BG experiments, we used the following annealing iterations:

$$\underbrace{1200\,\text{K}}_{T_1} \to \underbrace{1028.69\,\text{K}}_{T_2} \to 881.84\,\text{K} \to 755.95\,\text{K} \to 648.04\,\text{K} \to 555.52\,\text{K}$$
$$\to 476.22\,\text{K} \to 408.24\,\text{K} \to 349.96\,\text{K} \to \underbrace{300.0\,\text{K}}_{T_{K-1}} \to \underbrace{300.0\,\text{K}}_{T_K = T_\text{target}} \tag{13}$$

As described in the main text, the hexapeptide additionally used intermediate fine-tuning iterations after each annealing iteration, not only in the very end:

$$\underbrace{1200\,\text{K}}_{T_1} \to \underbrace{1028.69\,\text{K}}_{T_2} \to 1028.69\,\text{K} \to 881.84\,\text{K} \to 881.84\,\text{K} \to 755.95\,\text{K} \to 755.95\,\text{K}$$
$$\to 648.04\,\text{K} \to 648.04\,\text{K} \to 555.52\,\text{K} \to 555.52\,\text{K} \to 476.22\,\text{K} \to 476.22\,\text{K} \to 408.24\,\text{K} \to 408.24\,\text{K}$$
$$\to 349.96\,\text{K} \to 349.96\,\text{K} \to \underbrace{300.0\,\text{K}}_{T_{K-1}} \to \underbrace{300.0\,\text{K}}_{T_K = T_\text{target}} \tag{14}$$

As described in the main text, in each iteration of the annealing workflow, we use a buffered dataset for training, resampled according to the importance weights. Similarly to how we clipped the importance weights to calculate the forward KLD of the reweighted Ramachandrans, also here we clipped the highest 0.01 % of the importance weights to the lowest value among them. This can prevent overemphasizing outliers in the importance weights, though the effect is minimal.

We further note that instead of using only the training buffer dataset of the current annealing iteration, one can also reuse older samples that correspond to previous annealing iterations by reweighting them to the current target temperature. To simplify the workflow, we did not pursue this option in our experiments, but it might further increase sample efficiency.

### F.3. Ablation Experiments

In this section, we discuss several ablation experiments to analyze the impact of different hyperparameter choices. To keep the computational cost down, we performed each ablation experiment only once (except for the ablation experiments on starting temperatures). All ablation experiments start from the hyperparameters chosen in the main experiments, changing only the hyperparameters specified.

When ablating choices in the annealing workflow, we start all experiments from the same checkpoint pre-trained with reverse KLD at high temperature, to remove variations in the pre-training.

**Direct Importance Sampling** Figure 5 shows the result when directly performing importance sampling from 1200 K to 300 K, without using the iterative annealing workflow. As one can see, the overlap between the two distributions is very small, resulting in a very noisy result due to the small effective sample size. This shows that a multistep annealing workflow is necessary.

**Starting Temperature** As discussed in the main text, we perform high-temperature reverse KLD training in the beginning of our workflow to pre-train the flow distribution. The starting temperature $T_1$ at which reverse KLD training is performed needs to be chosen high enough such that minima are less separated and mode collapse is avoided.

In Figure 6, we analyze the propensity for mode collapse for each of the three studied systems. For each temperature, we performed four reverse KLD experiments and display the fraction of experiments with mode collapse among them (mode collapse is defined here by manual visual inspection of the Ramachandran plots). As one can see, for each system, a critical temperature exists above which mode collapse does not occur.

We note that the propensity for mode collapse not only depends on the temperature, but also on the chosen batch size and the number of gradient descent steps. The experiments in Figure 6 were performed with a batch size of 1024 and 250 000 gradient descent steps. Generally, when increasing the target temperature beyond the so-obtained critical temperature, one can use a smaller batch size and fewer gradient descent steps without obtaining mode collapse. This allows more efficient training in terms of target energy evaluations. For our main experiments, we thus chose a relatively large initial temperature

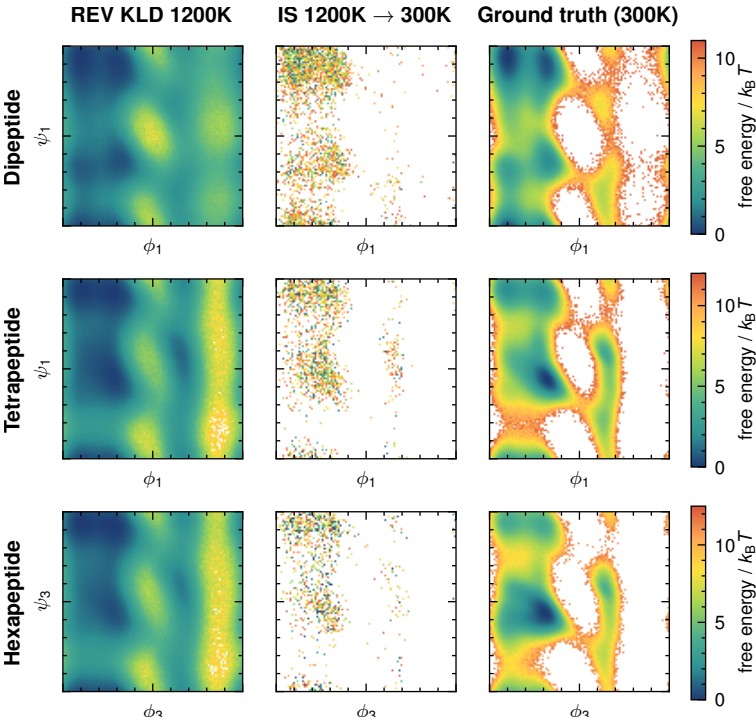

*Figure 5.* Free energy $F = -k_\mathrm{B}T \ln p(\phi_i, \psi_i)$ of dihedral angles (Ramachandran plots), reweighted directly from 1200 K to 300 K. We used $1 \times 10^6$ samples for importance sampling (middle).

$T_1 = 1200\,\mathrm{K}$. A tradeoff exists: Increasing $T_1$ allows cheaper reverse KLD training without mode collapse, but also requires more annealing steps to reach the desired target temperature.

**Temperature Schedule** As discussed in Section 4, we used a geometric progression between $T_\mathrm{start}$ and $T_\mathrm{target}$ as the temperature schedule for our main experiments. We compare this choice with a linear temperature schedule for alanine dipeptide in Figure 7, and for the hexapeptide in Figure 8. As one can see, the geometric temperature schedule is able to keep an approximately constant buffer ESS and therefore overlap between two consecutive distributions, while the buffer ESS drops down significantly for the linear schedule. In the future, also adaptive schedules can be explored (Goshtasbpour et al., 2023) to further improve the transition from one temperature level to the next.

**Final Fine-Tuning** As described in Section 4, all TA-BG experiments used a final fine-tuning iteration with $T_{K-1} = T_K$. Importance sampling to the same temperature (without lowering the target temperature) yields higher overlap between the two distributions, as is evident from the increased buffer ESS in the final iteration shown in Figure 7. We found empirically that such a final fine-tuning iteration with increased buffer ESS is helpful to improve the final metrics for all systems, as is summarized in Table 6.

**Intermediate Fine-Tuning** For the hexapeptide system, we additionally added intermediate fine-tuning iterations after each annealing iteration, instead of only in the very end. We compare the impact of such intermediate fine-tuning for all systems in Table 7. As we can see, the impact is the largest for the hexapeptide system, where including intermediate fine-tuning iterations significantly boosts the final ESS (see also Figure 9). Since such intermediate fine-tuning costs additional target energy evaluations, we did not include them for the two smaller systems.

**NO. Annealing Iterations** For our main experiments, we annealed the temperature 9 times for all systems. We summarize the impact of the number of annealing iterations on the final metrics for alanine dipeptide in Table 8. Choosing the number of annealing iterations is a tradeoff between improving the final metrics and lowering the number of target evaluations.

**NO. Samples Per Annealing Iteration**    Also the number of samples drawn to form the training buffer datasets in each annealing iteration forms a tradeoff between improving the final metrics and lowering the number of target evaluations. This is summarized in Table 9 for alanine dipeptide, where we draw the specified number of samples and resample this dataset to the same size for training in each annealing iteration.

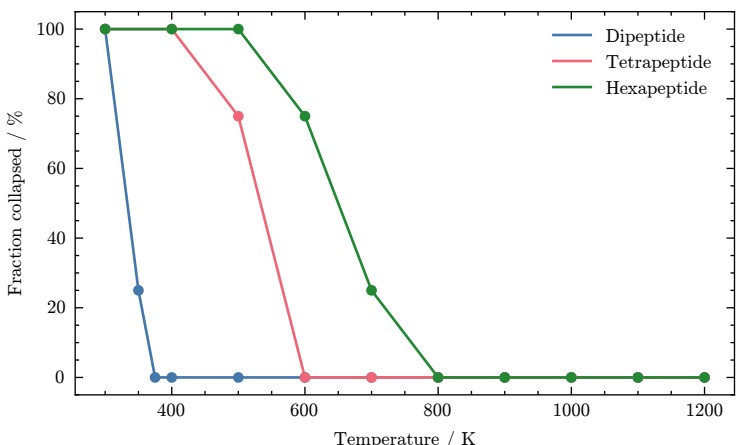

*Figure 6.* Ablation of starting temperature $T_1$: Fraction of reverse KLD experiments with mode collapse at a given temperature.

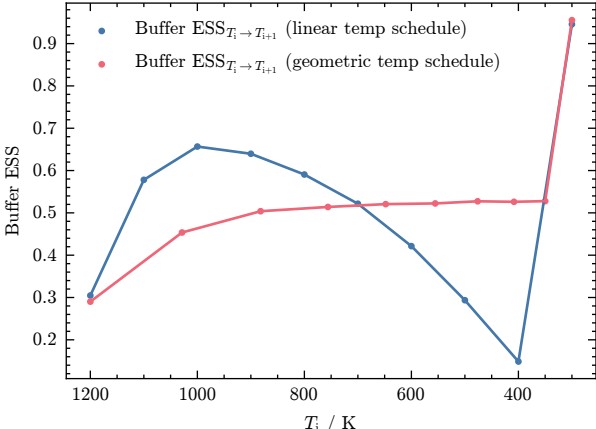

*Figure 7.* The ESS of the training buffer datasets $\mathcal{W}$ used to anneal from $T_i$ to $T_{i+1}$, comparing a linear and geometric temperature schedule for alanine dipeptide. The increase of the buffer ESS in the end is due to the final fine-tuning iteration with $T_{i+1} = T_i$.

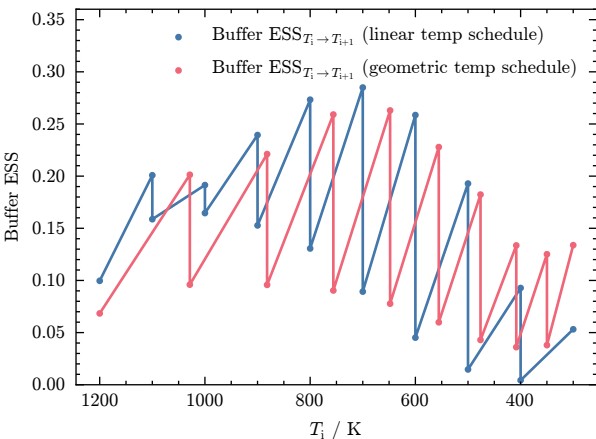

*Figure 8.* The ESS of the training buffer datasets $\mathcal{W}$ used to anneal from $T_i$ to $T_{i+1}$, comparing a linear and geometric temperature schedule for alanine hexapeptide. Note that the hexapeptide system includes intermediate fine-tuning iterations ($T_{i+1} = T_i$), where the buffer ESS is increased due to better overlap compared to the iterations where the temperature is lowered ($T_{i+1} < T_i$).

*Table 6.* Impact of final fine-tuning step on final metrics per system.

| SYSTEM | INTERMED. FINE-TUNING | FINAL FINE-TUNING | NLL | ESS / % |
|---|---|---|---|---|
| ALANINE DIPEPTIDE | WITHOUT | WITH | -213.667 | 95.85 |
| | WITHOUT | WITHOUT | -213.666 | 95.55 |
| ALANINE TETRAPEPTIDE | WITHOUT | WITH | -330.128 | 62.57 |
| | WITHOUT | WITHOUT | -330.087 | 57.57 |
| ALANINE HEXAPEPTIDE | WITH | WITH | -501.510 | 15.38 |
| | WITH | WITHOUT | -501.461 | 13.37 |

*Table 7.* Impact of intermediate fine-tuning on final metrics per system.

| SYSTEM | INTERMED. FINE-TUNING | NLL | ESS / % |
|---|---|---|---|
| ALANINE DIPEPTIDE | WITH | -213.671 | 96.46 |
| | WITHOUT | -213.667 | 95.85 |
| ALANINE TETRAPEPTIDE | WITH | -330.176 | 68.93 |
| | WITHOUT | -330.128 | 62.57 |
| ALANINE HEXAPEPTIDE | WITH | -501.510 | 15.38 |
| | WITHOUT | -499.580 | 7.56 |

*Table 8.* Impact of number of annealing iterations on final metrics for alanine dipeptide.

| NO. ANNEALING ITERATIONS | NLL | ESS / % |
|---|---|---|
| 3 | -213.647 | 92.27 |
| 5 | -213.663 | 95.06 |
| 7 | -213.666 | 95.57 |
| 9 | -213.667 | 95.85 |
| 11 | -213.668 | 96.05 |

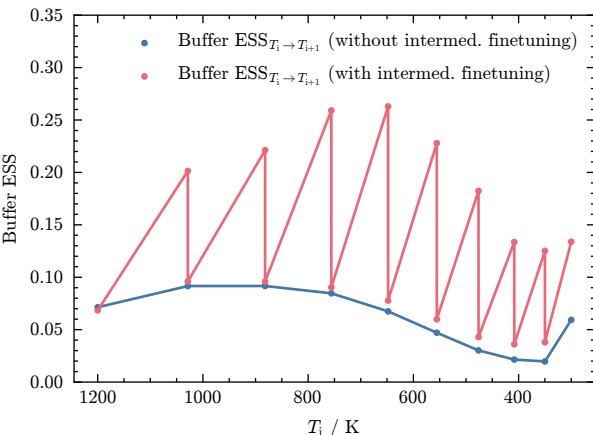

*Figure 9.* The ESS of the training buffer datasets $\mathcal{W}$ used to anneal from $T_i$ to $T_{i+1}$, comparing the case with and without intermediate fine-tuning iterations for the hexapeptide system.

*Table 9.* Impact of number of samples drawn per annealing iteration on final metrics for alanine dipeptide.

| NO. SAMPLES | NLL | ESS / % |
|---|---|---|
| 500 000 | -213.556 | 84.49 |
| 1 000 000 | -213.649 | 92.47 |
| 2 000 000 | -213.663 | 95.12 |
| 5 000 000 | -213.670 | 96.43 |
| 10 000 000 | -213.673 | 96.86 |

## F.4. Scaling to Higher Dimensions

Our proposed annealing workflow relies on importance sampling (IS) to anneal the distribution iteratively from $T_i$ to $T_{i+1}$. Since importance sampling does not scale well when increasing the dimensionality $N$ of the problem, we want to shortly discuss how annealed importance sampling (AIS) (Neal, 2001) can be used in place of IS in the future, and how this can mitigate potential problems encountered with IS when scaling to larger systems.

The main problem of IS is that small local bias, e.g., the possibility of two atoms clashing locally in the proposal, gets amplified due to the dimensionality of the problem. We consider a molecular system with $N$ independent neighborhoods, each with a chance $\eta$ for a clash under the proposal distribution. A sample's importance weight is $1$ if no clashes occurred, and $0$ otherwise. The ESS is $\text{ESS} = \frac{\left(\sum_{i=1}^{M} w_i\right)^2}{M \sum_{i=1}^{M} w_i^2} = \frac{1}{M} \sum_{i=1}^{M} w_i$, so $\mathbb{E}(\text{ESS}) = (1 - \eta)^N$. Thus, the ESS drops exponentially, which captures the scaling problem of IS.

To demonstrate how AIS addresses this, we move to a continuous model (a similar example and analysis can be found in (Midgley et al., 2023b)): The proposal distribution is an $N$-dimensional Gaussian with $\sigma = 1.1$, $\mu = 0$, and the target distribution is an $N$-dimensional Gaussian with $\sigma = 1.0$, $\mu = 0$. This is a simplified model of a single annealing iteration. To perform AIS in this toy example, we use a single step of HMC with 5 leap-frog steps to transition between two intermediate distributions. We further scale the number of intermediate distributions $T$ linearly with the dimensionality $N$, we chose $T = 5 \cdot N$ in this example.

We visualize the results of this toy example in Figure 10. As one can see, the ESS of vanilla IS drops exponentially. However, when using AIS and scaling the number of intermediate distributions linearly with $N$, one can obtain an approximately constant ESS. While this is, of course, a simplified setup, it still captures the essence of why IS fails for higher dimensions and how AIS can mitigate this issue. We also refer to (Neal, 2001) for a theoretical analysis. While for molecular systems, exact factorizability of the distribution is of course not given, AIS can still remove atom clashes, etc., which are often local effects. Thus, by using AIS instead of IS, potentially arising scaling problems can be avoided in the future.

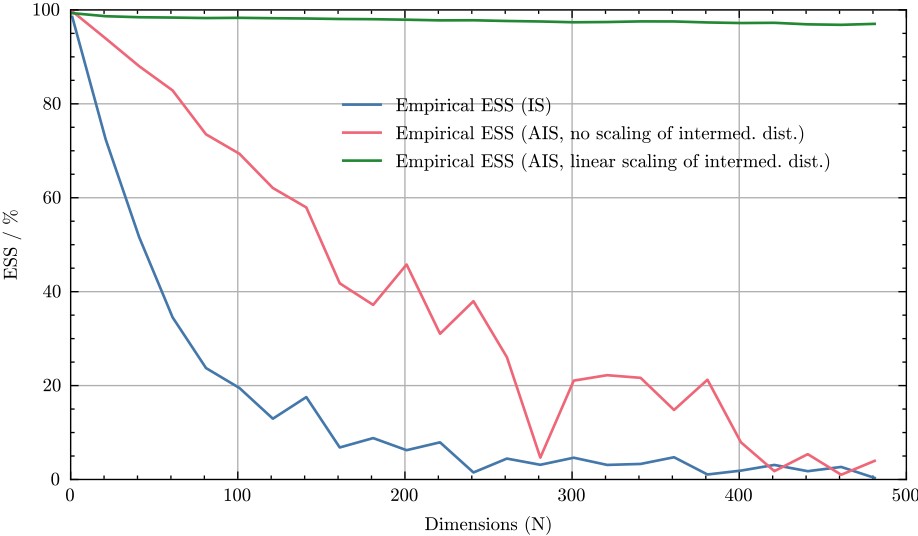

*Figure 10.* Comparison of the empirical ESS as a function of the number of dimensions $N$ for a Gaussian toy system, as obtained using vanilla IS, AIS without increasing the number of intermediate distributions, and AIS when linearly scaling up the number of intermediate distributions with $N$.

# G. FAB

## G.1. Hyperparameters of Main Experiments

For the FAB experiments, we started from the hyperparameters reported in the original publication. To make the original FAB hyperparameters a good starting point, we scaled our internal coordinates from the range of the splines [0,1] to [0,10], which is the range used in the original FAB implementation. With this, we used an initial HMC step size of $0.05$ for all experiments. Since FAB obtained significantly better results by using a prioritized replay buffer, and we were able to reproduce this finding, we chose the same replay buffer used originally by FAB for all experiments.

*Table 10.* Hyperparameters of the FAB experiments at $300\,\mathrm{K}$. All experiments used a cosine annealing learning rate scheduler with a single cycle.

|  | DIPEPTIDE | TETRAPEPTIDE | HEXAPEPTIDE |
|---|---|---|---|
| GRADIENT DESCENT STEPS | 50 000 | 50 000 | 50 000 |
| LEARNING RATE | $1 \times 10^{-4}$ | $1 \times 10^{-4}$ | $1 \times 10^{-4}$ |
| BATCH SIZE | 1024 | 1024 | 1024 |
| GRAD NORM CLIPPING | 1000.0 | 1000.0 | 1000.0 |
| LR LINEAR WARMUP STEPS | 1000 | 1000 | 1000 |
| WEIGHT DECAY (L2) | $1 \times 10^{-5}$ | $1 \times 10^{-5}$ | $1 \times 10^{-5}$ |
| NO. INTERMED. DIST. | 8 | 8 | 8 |
| NO. INNER HMC STEPS | 4 | 4 | 8 |

## G.2. Ablation Experiments

As with most sampling approaches, the number of target evaluations and the accuracy of the obtained distribution form a tradeoff. For the hexapeptide, FAB was not able to resolve the metastable high-energy region accurately. Therefore, we performed additional experiments where we varied the number of intermediate distributions and the number of HMC steps. This improves the results slightly, while requiring significantly more target energy evaluations (see Table 11).

For the smaller systems, alanine dipeptide and alanine tetrapeptide, we further performed experiments with smaller batch sizes while using the same number of gradient descent steps. This lowers the number of required target energy evaluations. However, as one can see from Table 11, this comes at the cost of further increasing the NLL.

*Table 11.* Impact of hyperparameter choices on the NLL of FAB. In bold are the hyperparameters that were used for the main experiments, chosen as a trade-off between computational cost and accuracy. Only the main experiments were performed four times, all other experiments were performed once.

| SYSTEM | BATCH SIZE | NO. INTERMED. DIST. | NO. INNER HMC STEPS | PE EVALS ↓ | NLL ↓ |
|---|---|---|---|---|---|
| ALANINE DIPEPTIDE | **1024** | **8** | **4** | $\mathbf{2.13 \times 10^8}$ | $\mathbf{-213.653 \pm 0.000}$ |
|  | 512 | 8 | 4 | $1.07 \times 10^8$ | $-213.643$ |
|  | 256 | 8 | 4 | $5.33 \times 10^7$ | $-213.623$ |
| ALANINE TETRAPEPTIDE | **1024** | **8** | **4** | $\mathbf{2.13 \times 10^8}$ | $\mathbf{-330.100 \pm 0.002}$ |
|  | 512 | 8 | 4 | $1.07 \times 10^8$ | $-330.019$ |
|  | 256 | 8 | 4 | $5.33 \times 10^7$ | $-329.874$ |
| ALANINE HEXAPEPTIDE | 1024 | 8 | 4 | $2.13 \times 10^8$ | $-501.157$ |
|  | 512 | 8 | 4 | $1.07 \times 10^8$ | $-500.857$ |
|  | 1024 | 16 | 4 | $4.20 \times 10^8$ | $-501.255$ |
|  | **1024** | **8** | **8** | $\mathbf{4.20 \times 10^8}$ | $\mathbf{-501.268 \pm 0.008}$ |
|  | 1024 | 16 | 8 | $8.34 \times 10^8$ | $-501.327$ |

# H. Reverse KLD

## H.1. Hyperparameters of Main Experiments

*Table 12.* Hyperparameters of the reverse KLD experiments. This includes the 300 K experiments, but also the 1200 K experiments used to start the temperature-annealing workflow. All experiments used a cosine annealing learning rate scheduler with a single cycle.

| $T$ | DIPEPTIDE 300 K | TETRAPEPTIDE 300 K | HEXAPEPTIDE 300 K | DIPEPTIDE 1200 K | TETRAPEPTIDE 1200 K | HEXAPEPTIDE 1200 K |
|---|---|---|---|---|---|---|
| GRADIENT DESCENT STEPS | 250 000 | 250 000 | 250 000 | 100 000 | 100 000 | 250 000 |
| LEARNING RATE | $1 \times 10^{-4}$ | $1 \times 10^{-4}$ | $1 \times 10^{-4}$ | $1 \times 10^{-4}$ | $1 \times 10^{-4}$ | $1 \times 10^{-4}$ |
| BATCH SIZE | 1024 | 1024 | 1024 | 256 | 256 | 512 |
| GRAD NORM CLIPPING | 100.0 | 100.0 | 100.0 | 100.0 | 100.0 | 100.0 |
| LR LINEAR WARMUP STEPS | 1000 | 1000 | 1000 | 1000 | 1000 | 1000 |
| WEIGHT DECAY (L2) | $1 \times 10^{-5}$ | $1 \times 10^{-5}$ | $1 \times 10^{-5}$ | $1 \times 10^{-5}$ | $1 \times 10^{-5}$ | $1 \times 10^{-5}$ |
| NO. HIGHEST ENERGY VALUES REMOVED | 40 | 40 | 40 | 10 | 10 | 20 |

# I. Forward KLD

## I.1. Hyperparameters of Main Experiments

*Table 13.* Hyperparameters of the forward KLD experiments at 300 K. All experiments used a cosine annealing learning rate scheduler with a single cycle.

| | DIPEPTIDE | TETRAPEPTIDE | HEXAPEPTIDE |
|---|---|---|---|
| GRADIENT DESCENT STEPS | 100 000 | 100 000 | 120 000 |
| LEARNING RATE | $5 \times 10^{-5}$ | $5 \times 10^{-5}$ | $5 \times 10^{-5}$ |
| BATCH SIZE | 1024 | 1024 | 1024 |

## J. Comparison of Computational Cost

In this section, we compare the computational cost of TA-BG and FAB, summarized in Table 14. We point out that neither of the two implementations was optimized for speed, thus the wall times are only approximately indicative of the actually achievable performance.

As already discussed, TA-BG is more efficient compared to FAB in terms of target energy evaluation. However, this increased sampling efficiency comes with the cost of more flow evaluations, since we train for an extended period of time on large buffer datasets. The force field evaluations of the benchmark systems in this work are relatively inexpensive. Thus, the total wall time of FAB is currently slightly lower compared to TA-BG for the systems investigated here.

However, we note that while the force field evaluations are inexpensive, they are also not very accurate. When applying our approach to systems with more expensive target evaluations, such as ones based on foundation model force fields or density functional theory, target evaluations become dominant, favoring methods with improved sampling efficiency, such as TA-BG.

*Table 14.* Computational cost of TA-BG and FAB, comparing the number of target evaluations, the number of flow evaluations (batched), and the total wall time (excluding evaluation). To determine the wall times, we ran 4 experiments in parallel on a compute node with $2 \times$ AMD EPYC Rome 7402 CPU (24 cores each) and $4 \times$ NVIDIA A100 GPU.

| SYSTEM | METHOD | NO. TARGET EVALS | NO. FLOW EVALS (BATCHES) | WALL TIME |
|---|---|---|---|---|
| ALANINE DIPEPTIDE | FAB | $2.13 \times 10^8$ | $2.06 \times 10^5$ (AIS UPDATE BUFFER, BS 1024) $+ 5 \times 10^4$ (TRAINING, BS 1024) $= 2.56 \times 10^5$ | 18.0 h |
| | TA-BG | $2.56 \times 10^7$ (REV. KLD PRE-TRAINING) $+ 5 \times 10^7$ (ANNEALING) $= 7.56 \times 10^7$ | $1 \times 10^5$ (REV. KLD PRE-TRAINING, BS 256) $+ 5 \times 10^7 / 4096$ (BUFFER CREATION, BS 4096) $+ 3 \times 10^5$ (FWD. KLD ANNEALING, BS 2048) $= 4.12 \times 10^5$ | 22.2 h |
| ALANINE TETRAPEPTIDE | FAB | $2.13 \times 10^8$ | $2.06 \times 10^5$ (AIS UPDATE BUFFER, BS 1024) $+ 5 \times 10^4$ (TRAINING, BS 1024) $= 2.56 \times 10^5$ | 19.67 h |
| | TA-BG | $2.56 \times 10^7$ (REV. KLD PRE-TRAINING) $+ 5 \times 10^7$ (ANNEALING) $= 7.56 \times 10^7$ | $1 \times 10^5$ (REV. KLD PRE-TRAINING, BS 256) $+ 5 \times 10^7 / 4096$ (BUFFER CREATION, BS 4096) $+ 2 \times 10^5$ (FWD. KLD ANNEALING, BS 4096) $= 3.12 \times 10^5$ | 23.3 h |
| ALANINE HEXAPEPTIDE | FAB | $4.20 \times 10^8$ | $4.06 \times 10^5$ (AIS UPDATE BUFFER, BS 1024) $+ 5 \times 10^4$ (TRAINING, BS 1024) $= 4.56 \times 10^5$ | 41.4 h |
| | TA-BG | $1.28 \times 10^8$ (REV. KLD PRE-TRAINING) $+ 1.8 \times 10^8$ (ANNEALING) $= 3.08 \times 10^8$ | $2.5 \times 10^5$ (REV. KLD PRE-TRAINING, BS 512) $+ 1.8 \times 10^8 / 4096$ (BUFFER CREATION, BS 4096) $+ 3.6 \times 10^5$ (FWD. KLD ANNEALING, BS 2048) $= 6.5 \times 10^5$ | 52.2 h |

## K. 2D GMM System

Next to the molecular systems, we additionally consider the 40 Gaussian mixture density in 2 dimensions as introduced by Midgley et al. (2023b). The ground truth distribution, as well as the results obtained with TA-BG and FAB, are visualized in Figure 11.

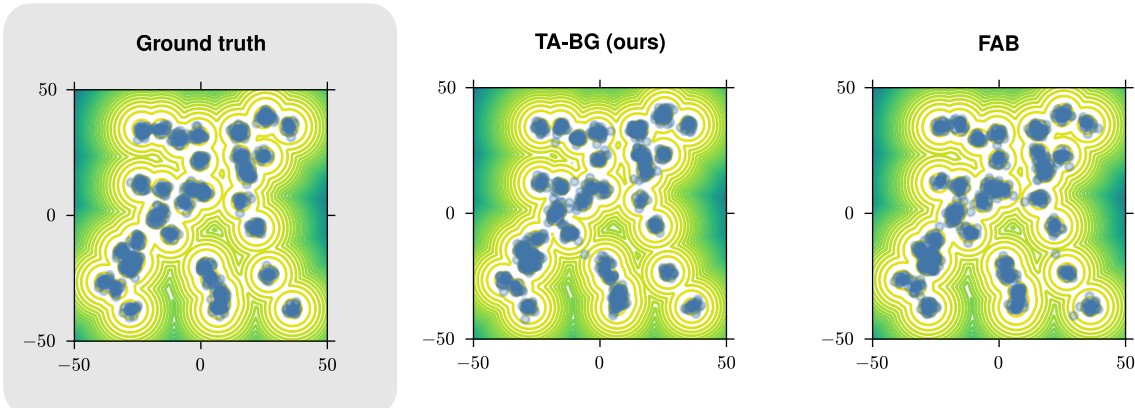

*Figure 11.* Comparison of obtained distributions for the 2D GMM system. We visualize 1000 samples from the ground truth distribution, TA-BG, and FAB. We additionally visualize the contour lines of the ground truth log probability.

We report the obtained NLL and ESS in Table 15. This table additionally contains results for diffusion-based samplers: Path Integral Sampler (PIS, Zhang & Chen (2021)), Denoising Diffusion Sampler (DDS, Vargas et al. (2022)), and Iterated Denoising Energy Matching (iDEM, Akhound-Sadegh et al. (2024)). For these methods, we show the results reported by Akhound-Sadegh et al. (2024).

FAB (REAL NVP) in Table 15 refers to the original FAB results reported by Akhound-Sadegh et al. (2024), which are based on the Real NVP coupling flow architecture (Dinh et al., 2017). We repeated experiments with FAB based on the more expressive neural splines coupling flows (Durkan et al., 2019), reported as FAB (NEURAL SPLINES) in Table 15. The same architecture was used for our TA-BG experiments.

As one can see both from Figure 11 and Table 15, TA-BG and FAB obtain strong results on the 2D GMM system, outperforming or matching the considered diffusion-based baselines. The results of TA-BG and FAB are practically identical and likely close to the optimal solution possible with the given architecture.

We note that the 2D GMM system is a very simple target, thus the inclusion here mostly serves illustrative purposes to show that our methodology can also be applied to more traditional sampling tasks. Since the informative value of a systematic comparison on this system is limited, we did not tune the results of TA-BG and FAB with respect to the number of target energy evaluations.

TA-BG can, of course, also be applied to other sampling tasks considered in related literature. However, we purposely set the focus of this work on molecular systems, where the potential energy is more correlated and complex compared to many of the toy problems currently considered in the variational sampling literature. Symmetric target densities with identical particles, such as DW-4, LJ-13, or LJ-55 (Akhound-Sadegh et al., 2024), should ideally be handled with an equivariant flow architecture, thus we did not consider them here.

### K.1. Details

Here, we specify hyperparameters and other details for TA-BG and FAB applied to the GMM system. We only specify hyperparameters that differ from the ones used in our main experiments on the molecular systems.

Contrary to the experiments on the molecular systems, we did not use a regularized energy function for the GMM system. Also, we did not clip importance weights when calculating the ESS. Following previous work, the ESS was calculated using 1000 samples from the model, the NLL was calculated using 1000 ground truth samples.

*Table 15.* Comparison of metrics on the 2D GMM system. The best-performing method for each metric is highlighted in bold. For our experiments (FAB (NEURAL SPLINES) and TA-BG (NEURAL SPLINES)), we specify the mean and standard deviation of the metrics across four independent runs, the rest is reported with the mean and standard deviation over three independent runs.

| METHOD | NLL $\downarrow$ | ESS $\uparrow$ |
|---|---|---|
| FAB (REAL NVP) | $7.14 \pm 0.01$ | $(65.3 \pm 1.7)\,\%$ |
| FAB (NEURAL SPLINES) | $\mathbf{6.92 \pm 0.00}$ | $\mathbf{(97.09 \pm 0.58)\,\%}$ |
| TA-BG (NEURAL SPLINES) | $\mathbf{6.91 \pm 0.01}$ | $\mathbf{(96.89 \pm 0.48)\,\%}$ |
| PIS | $7.72 \pm 0.03$ | $(29.5 \pm 1.8)\,\%$ |
| DDS | $7.43 \pm 0.46$ | $(68.7 \pm 20.8)\,\%$ |
| IDEM | $\mathbf{6.96 \pm 0.07}$ | $(73.4 \pm 9.2)\,\%$ |

**Architecture** For all experiments on the GMM system, we used monotonically increasing rational-quadratic splines with $K = 16$ bins in the range $[-50, 50]$. For the parameter networks, we used fully connected neural networks with hidden dimensions $[120, 120]$ and ReLU activation functions. We used 13 coupling layers, swapping the two dimensions after each coupling. As the latent distribution $q_Z$, we used a 2D normal distribution with $\mu = 0$ and $\sigma = 10.0$, truncated to the range $[-50, 50]$.

**FAB Hyperparameters** We used a learning rate of $1 \times 10^{-5}$, a batch size of $8192$, and $50\,000$ gradient descent steps. We scaled the coordinates from $[-50, 50]$ to $[-5, 5]$ to perform AIS. Contrary to the experiments on the molecular systems, we set both the number of intermediate AIS distributions and the number of inner HMC steps to one. We further used $0.05$ as the initial HMC step size.

**TA-BG Hyperparameters** Reverse KLD pre-training was performed at $T = 30.0$, using a learning rate of $1 \times 10^{-4}$, a batch size of $128$, and $50\,000$ gradient descent steps. Contrary to the reverse KLD experiments on the molecular systems, we here did not remove the largest energy values in the loss contributions of each batch. We also did not use a learning rate scheduler.

For the annealing, we used a learning rate of $5 \times 10^{-5}$, a batch size of $8192$, and no learning rate scheduler. We used the following annealing iterations, following a geometric temperature schedule:

$$\underbrace{30.0\,\text{K}}_{T_1} \to \underbrace{18.45\,\text{K}}_{T_2} \to 11.35\,\text{K} \to 6.98\,\text{K} \to 4.30\,\text{K} \to 2.64\,\text{K} \to 1.63\,\text{K} \to \underbrace{1.0\,\text{K}}_{T_{K-1}} \to \underbrace{1.0\,\text{K}}_{T_K = T_{\text{target}}} \tag{15}$$

For each annealing iteration, we draw $2\,000\,000$ samples, resample to $2\,000\,000$ samples using the importance weights (without clipping), and perform $20\,000$ forward KLD gradient descent steps.

# L. Additional Figures

## L.1. All Systems (Reweighted)

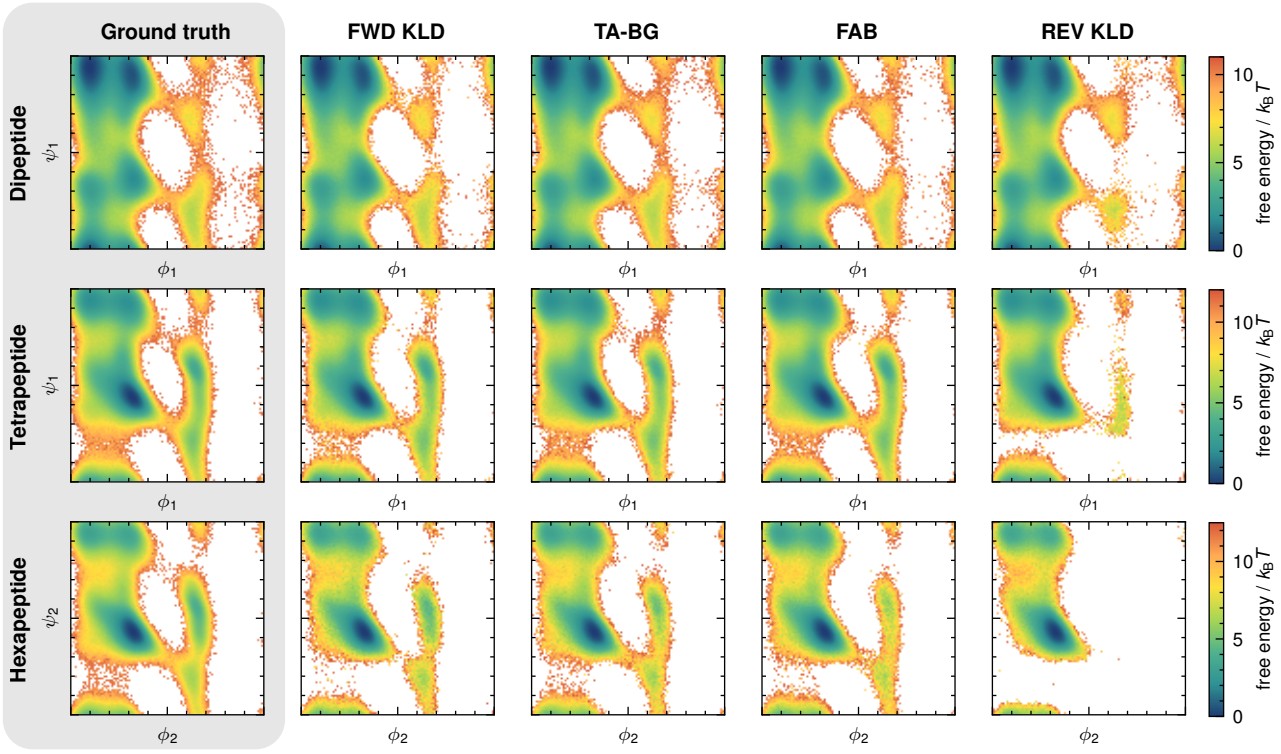

*Figure 12.* Reweighted version of Figure 3. Comparison of the free energy $F = -k_{\mathrm{B}}T \ln p(\phi_i, \psi_i)$ of selected dihedral angles (Ramachandran plots) at 300 K, reweighted to 300 K.

## L.2. Alanine Tetrapeptide

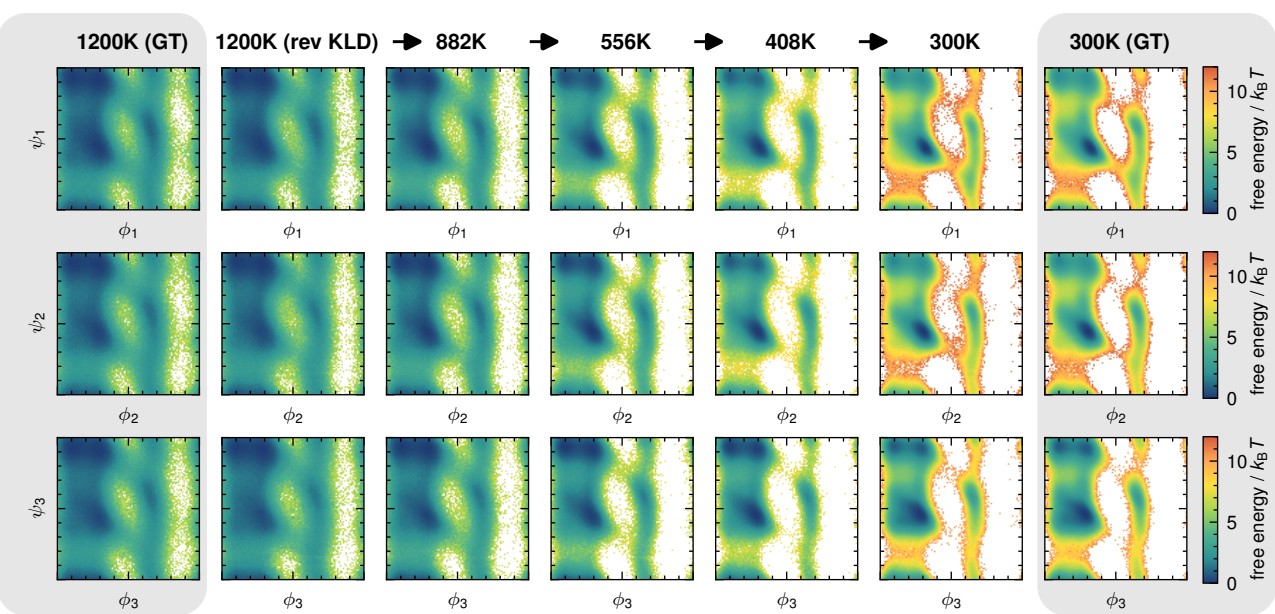

*Figure 13.* Visualization of the iterative annealing process for the tetrapeptide, showing the free energy $F = -k_\mathrm{B}T \ln p(\phi_i, \psi_i)$ of backbone dihedral angles (Ramachandran plots) in each iteration. Note that not all annealing iterations are shown. We used $1 \times 10^7$ samples for the Ramachandran plots at 300 K and $1 \times 10^6$ samples for the rest.

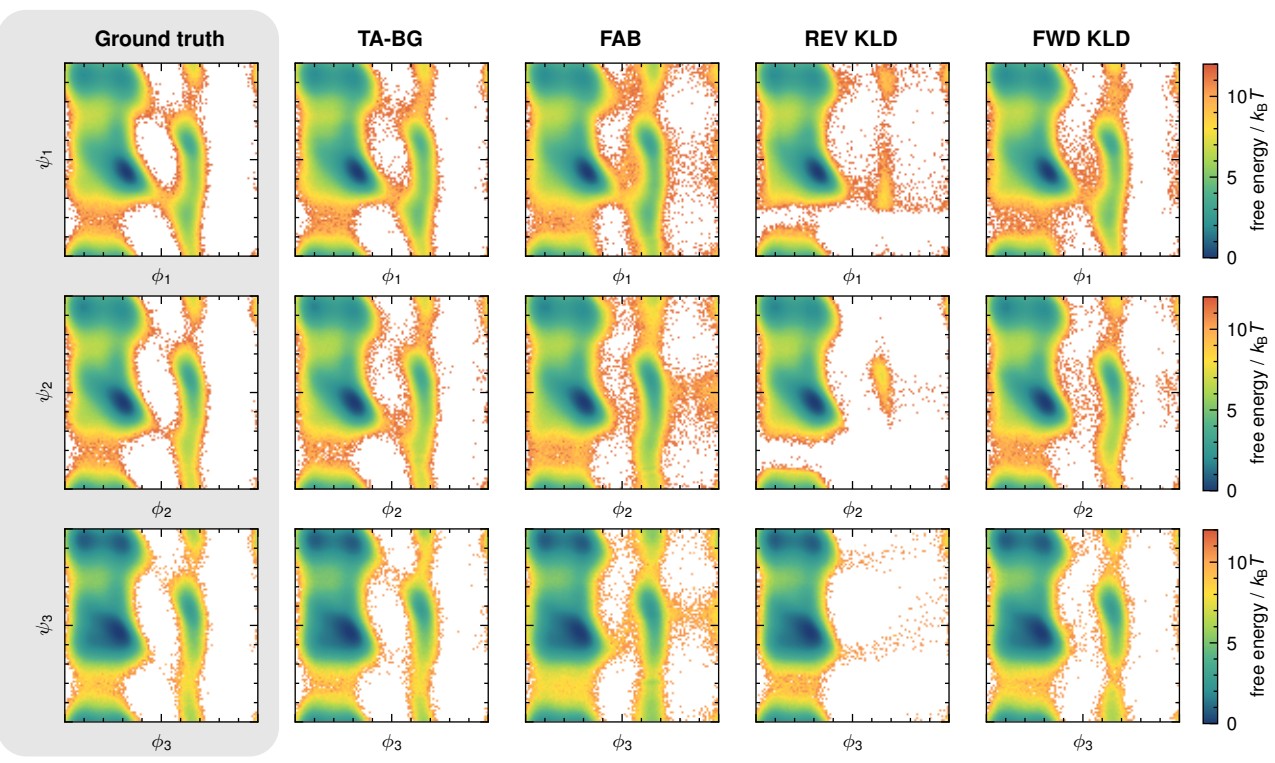

*Figure 14.* Comparison of the free energy $F = -k_{\mathrm{B}}T \ln p(\phi_i, \psi_i)$ of the backbone dihedral angles (Ramachandran plots) of the tetrapeptide at 300 K.

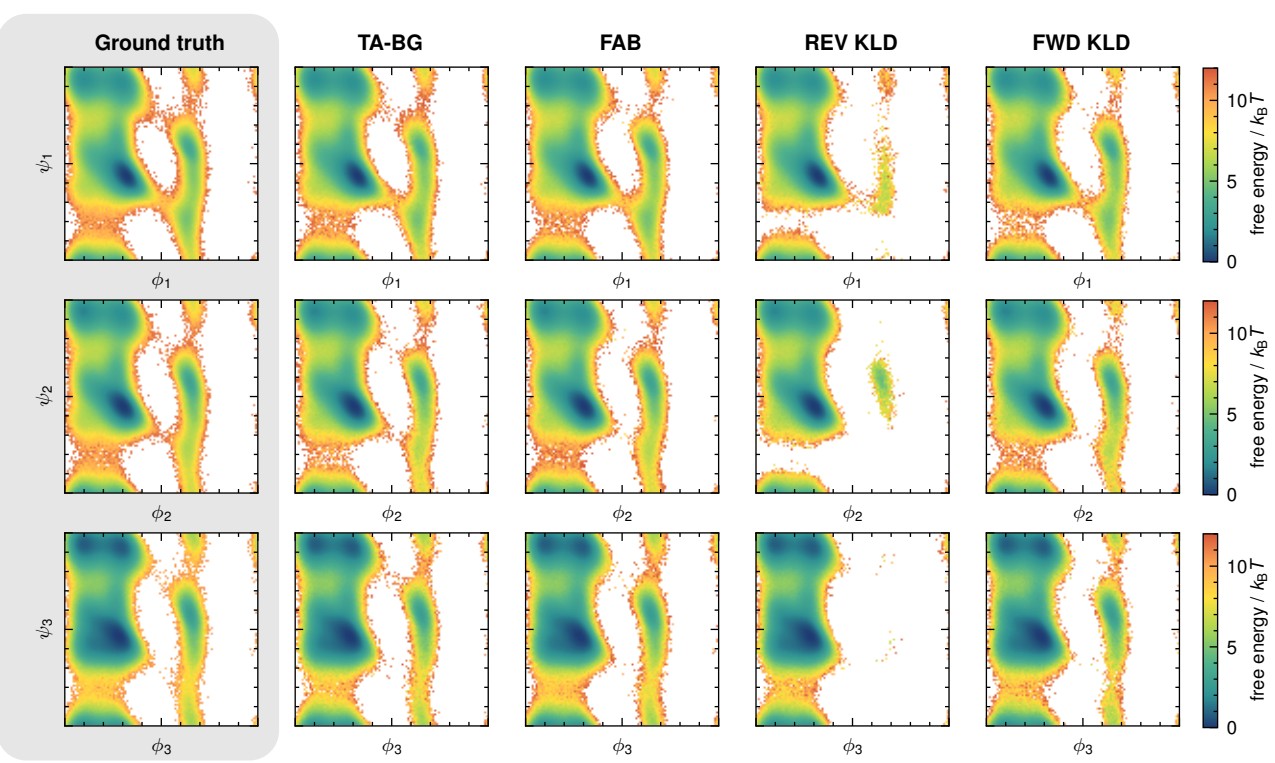

*Figure 15.* Reweighted version of Figure 14. Comparison of the free energy $F = -k_{\mathrm{B}} T \ln p(\phi_i, \psi_i)$ of the backbone dihedral angles (Ramachandran plots) of the tetrapeptide at 300 K, reweighted to 300 K.

## L.3. Alanine Hexapeptide

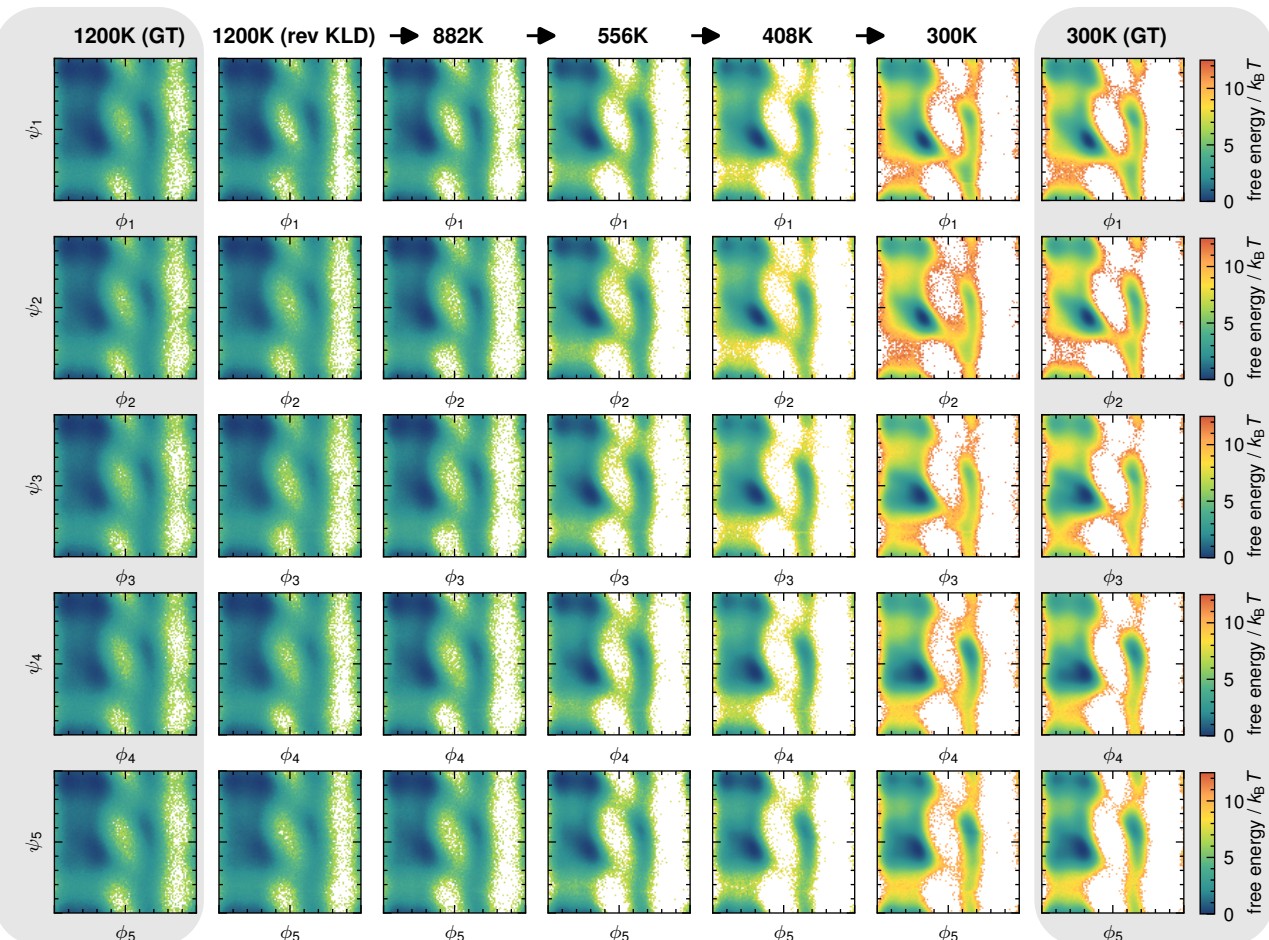

*Figure 16.* Visualization of the iterative annealing process for the hexapeptide, showing the free energy $F = -k_{\mathrm{B}}T \ln p(\phi_i, \psi_i)$ of backbone dihedral angles (Ramachandran plots) in each iteration. Note that not all annealing iterations are shown. We used $1 \times 10^7$ samples for the Ramachandran plots at 300 K and $1 \times 10^6$ samples for the rest.

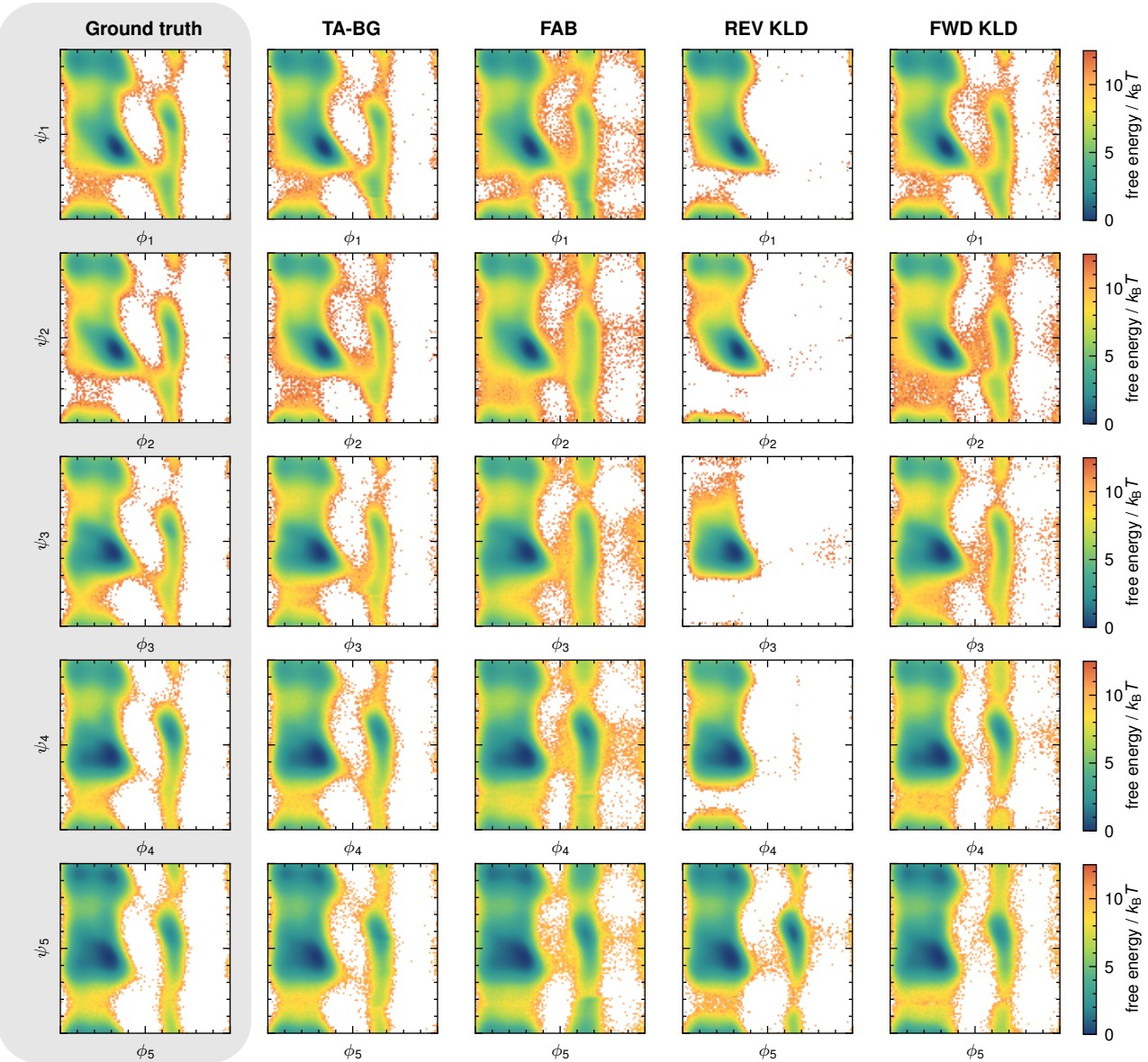

*Figure 17.* Comparison of the free energy $F = -k_{\mathrm{B}}T \ln p(\phi_i, \psi_i)$ of the backbone dihedral angles (Ramachandran plots) of the hexapeptide at 300 K.

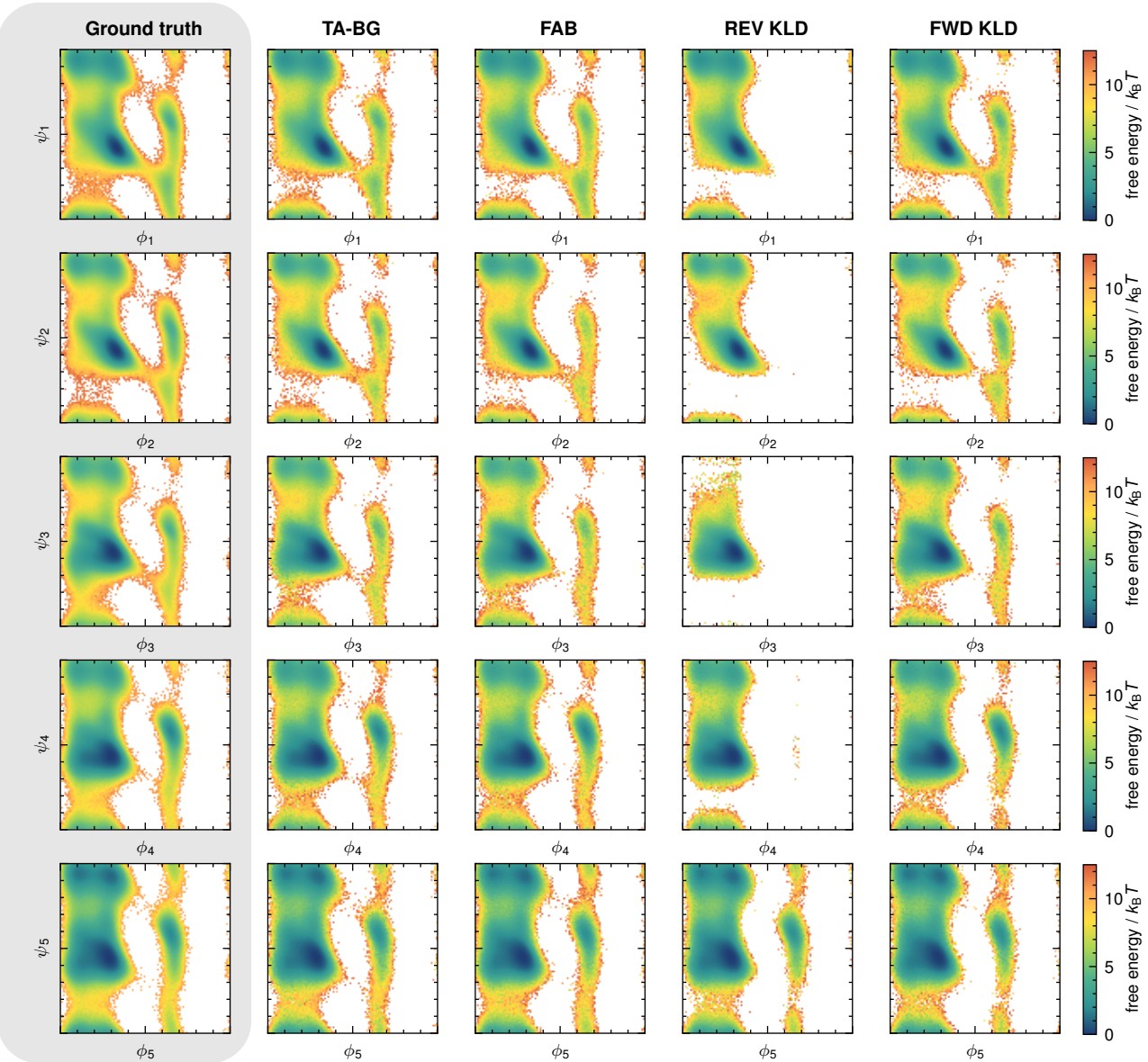

*Figure 18.* Reweighted version of Figure 17. Comparison of the free energy $F = -k_{\mathrm{B}}T \ln p(\phi_i, \psi_i)$ of the backbone dihedral angles (Ramachandran plots) of the hexapeptide at 300 K, reweighted to 300 K.

