# OpenReview forum: "Temperature-Annealed Boltzmann Generators"
_ICML.cc/2025/Conference — ICML 2025 poster_

### Official Review · Reviewer_BtBN · 2025-03-03

**Overall Recommendation:** 4

**Summary:**

In this paper, the authors present a temperature-annealing strategy to train normalizing flows to match unnormalized probability densities (in this case focusing on the equilibrium Boltzmann distribution of high-dimensional molecular systems). The training is done with the reverse KL divergence, assuming no access to any samples from the true density, and only the true energy function. To avoid the mode collapse issue associated with training with the reverse KL divergence objective, the authors propose to first train at a high temperature at which the sample density is closer to a uniform distribution and energy barriers are less difficult to overcome, followed by iterative reweighting and retraining at slightly lower temperatures, until the target temperature (e.g room temperature) is reached. The proposed method is demonstrated on three small protein systems, increasing in complexity from alanine dipeptide to alanine hexapeptide. The method avoids mode collapse in larger systems and successfully reproduces the ground truth distributions obtained from MD simulations.

**Claims And Evidence:**

The authors claim that their proposed annealing approach prevents mode collapse by starting at a higher temperature at which the sample density is closer to uniform, followed by gradual reweighting. This claim is generally supported by the results in Figures 2, 3 and Table 1, which shows that training with reverse KL without temperature annealing often results in mode collapse.

The results of the proposed TA-BG approach are only marginally better than the strongest baseline, FAB, with most of the improvement appearing in the number of potential energy evaluations (typically much lower than FAB). Can the authors discuss more in depth what the aspect of their method is that enables them to learn more efficiently in this way? This wasn’t particularly intuitive to me.

**Essential References Not Discussed:**

While not directly related since it uses data to learn generative models, the authors should consider citing this recent paper: Lewis, Sarah, et al. "Scalable emulation of protein equilibrium ensembles with generative deep learning." bioRxiv (2024): 2024-12.

**Experimental Designs Or Analyses:**

See above.

**Methods And Evaluation Criteria:**

I am concerned about the scalability of the approach to higher-dimensional systems. Specifically, I noted that the effective sample size (ESS) resulting from reweighting quickly becomes smaller (from 95% to 15%) as the system complexity increases from alanine dipeptide to alanine hexapeptide. This makes intuitive sense, as the overlaps between distributions become smaller in higher dimensions. As the authors correctly note, importance sampling between these distributions can become ineffective. I would like the authors to demonstrate, either empirically or theoretically, some scaling properties of their proposed method, perhaps as a function of system size and/or number/spacing of reweighting temperatures. I can suggest the fast-folding proteins from D.E. Shaw simulations [1] as a natural testbed for some slightly larger proteins with interesting conformational dynamics and metastable states.

[1] Lindorff-Larsen, Kresten, et al. "How fast-folding proteins fold." Science 334.6055 (2011): 517-520.

**Other Comments Or Suggestions:**

See above.

**Other Strengths And Weaknesses:**

Strength: The paper is very well-written and easy to follow.

**Questions For Authors:**

Is the reported ESS metric the result of reweighting from the original temperature (1200K) to the final temperature (300K), or from some combination of all of the intermediate steps?

**Relation To Broader Scientific Literature:**

As acknowledged by the authors, the idea of temperature annealing is similar to replica-exchange molecular dynamics, in which higher temperatures are used to circumvent high energy barriers and speed up sampling.

In relation to other variational sampling methods, the paper compares most closely with Flow Annealed Importance Sampling Bootstrap (FAB). The authors demonstrate that their method matches the ground truth distribution more closely than FAB for the systems considered, and requires fewer potential energy evaluations.

Regarding other generative modeling methods for molecular systems, the main other line of work is training deep, generative models given large-scale data from, e.g. equilibrium MD simulations [1, 2]. The main benefit of the presented method is that no ground truth data, which can be expensive to collect, is needed. However, I would like a more detailed justification from the authors as to why they believe their approach can remain relevant compared to this line of work, particularly as the availability of structural and simulation databases like the PDB and ATLAS [3] steadily grow. Relatedly, can the authors’ method make use of this growing data availability to improve the efficiency of learning with their method on new/unseen systems? I think some discussion regarding this would be useful to better situate the contribution in the current landscape of protein generative modeling.

[1] Lewis, Sarah, et al. "Scalable emulation of protein equilibrium ensembles with generative deep learning." bioRxiv (2024): 2024-12.

[2] Zheng, Shuxin, et al. "Predicting equilibrium distributions for molecular systems with deep learning." Nature Machine Intelligence 6.5 (2024): 558-567

[3] Vander Meersche, Yann, et al. "ATLAS: protein flexibility description from atomistic molecular dynamics simulations." Nucleic acids research 52.D1 (2024): D384-D392.

**Theoretical Claims:**

N/A

---

> ### Author Rebuttal · Authors · 2025-03-31
>
> Thank you very much for your detailed and helpful feedback. We will address your points in the following:
> * Comparison to FAB in accuracy and metrics: While Table 1 suggests TA-BG mainly improves in terms of the number of target evaluations, it’s important to note that the high-energy metastable regions constitute only a small part of the state space. Small NLL differences can thus have significant implications. For example, in the hexapeptide system, TA-BG and FAB show similar NLLs, but Ramachandran plots (Fig. 3) reveal that FAB fails to resolve the high-energy metastable region, while TA-BG succeeds. Therefore, quantitative metrics should be complemented with system-specific analyses, such as Ramachandran plots.
> * Comparison to FAB in computational cost: We want to briefly discuss why our method is substantially more efficient than FAB. FAB uses annealed importance sampling (AIS) with intermediate Hamiltonian Monte Carlo (HMC) transitions to evaluate the α=2 divergence loss. While this yields accurate results (except for the hexapeptide), the AIS costs a significant amount of potential energy evaluations. Our method simply uses the reverse KLD at high temperature, which is significantly less costly to evaluate. While the annealing adds some additional cost, it is still very efficient since large buffers are used, where training samples are reused several times. Therefore, in total, our method is significantly cheaper than FAB.
> * Scaling to larger systems: First, we would like to emphasize that scaling to the hexapeptide is already a big achievement in itself, as the previous SOTA method FAB was the only method that worked even on the smallest of our three systems studied. Despite this, we are confident that our method scales to larger systems. While it is true that the ESS drops down for the larger systems, this can be counteracted with a more expressive architecture, e.g., the one recently proposed in [3]. At the same time, even with the current flow architecture, one can counteract the drop in ESS with other measures:
>     * Increase in the number of temperature steps or the number of drawn samples per annealing step, see our ablations in the answer to reviewer SzWG.
>     * Additional fine-tuning steps, see our ablations in the answer to reviewer 1DKq.
>     * One can use AIS instead of IS to estimate the loss in the annealing steps. While this comes at an additional cost, it can keep the variance of the loss low when increasing the intermediate AIS steps linearly with the dimensionality (see [4] for a theoretical analysis).
> * Your question about data-driven vs data-free sampling of Boltzmann distributions is very interesting. First, we want to point out that the task of sampling the equilibrium Boltzmann distribution of proteins is significantly harder than just predicting folded structures, as is done in AlphaFold. While the references [1] and [2] that you cited tackle the task of equilibrium sampling, the results are only rough approximations of the true equilibrium distribution. Even though large MD datasets have become available recently, we believe that much more data is necessary to achieve good transferability that can replace direct sampling through MD or variational methods, such as the one proposed in our work.
> Another recent publication [5] trained transferable Boltzmann generators on a large dataset of dipeptide MD simulations, which is a very narrow chemical space. While the transferability was successfully shown, also here the transfer to new systems only resulted in a rough estimate of the true equilibrium distribution.
> We thus believe that a hybrid data / variational approach could be pursued in the future. In our method, MD data can be used to supplement the high-temperature pre-training, which will make finding new modes easier and may lead to quicker convergence. We agree that this discussion is very important and we will include a discussion of these aspects in our revised manuscript.
> * The reported ESS in Table 1 is the ESS when sampling from the flow distribution (which has been annealed through multiple iterations to 300K) in the very end, reweighting to the 300K target.
>
> We further note that we have now performed several new ablation studies, as detailed in our responses to reviewers SzWG and 1DKq. Lastly, we now additionally benchmarked against the 2D GMM system (see our rebuttal to reviewer CidM), where TA-BG outperforms FAB and all diffusion-based baselines.
>
> With this, we hope you agree with us that our work is a significant achievement that the sampling community can build upon in the future. We hope to see more methods applied to the complex benchmark systems proposed in our work, while we are eager to scale our method to even larger systems next.
>
> [3] Zhai et al. 2024, "Normalizing Flows are Capable Generative Models"
> [4] Midgley et al. 2023, “Flow Annealed Importance Sampling Bootstrap”
> [5] Klein et al. 2024, “Transferable Boltzmann Generators”

---

> > ### Comment · Reviewer_BtBN · 2025-04-01
> >
> > I thank the authors for their reply. Overall I think this is good work and would be of interest at ICML even in its current form. I would raise my score to a 4 if I saw more convincing empirical evidence of the scaling potential of this method, perhaps demonstrating AIS on the considered peptide systems. See specific comments/questions below.
> >
> > 1. The manner in which samples are reused from one annealing iteration to the next is still unclear to me and I didn't find an explanation in the paper or rebuttal. Is there are pre-specified temperature difference cutoff beyond which samples from the flow at one temperature are no longer reweighted to another temperature? On average, what is the fraction of samples that are reused compared to re-sampling?
> >
> > 2. Regarding the claims of scaling, I see from the ablations that choosing more steps for annealing or performing finetuning after each step helps improve the ESS for larger systems. However, I'm not sure this wouldn't run into some of the same issues as, e.g. free energy calculations, which also must perform many intermediate finetuning steps to maintain reasonable ESS at significant computational cost [1]. The rebuttal mentions that AIS can yield linear scaling of intermediate steps w.r.t. system size. That is very interesting, and I wonder if the authors could demonstrate a small example/teaser of AIS in this context and show it's better scaling properties?
> >
> > 3. I find the claim of using more expressive architectures to solve this problem slightly dubious, as fundamentally the reweighting problem is a property of the underlying Boltzmann distributions, and not the learnable function class used to approximate them. Please let me know if I'm missing something.
> >
> > [1] "Free energies at QM accuracy from force fields via multimap targeted estimation", PNAS 2023
> >
> > [EDIT]: In light of the AIS analysis provided by the authors in the response below, I am willing to increase my score from 3 -> 4. I appreciate this analysis, and I would like to see it included in the final paper. [Minor detail] I would also like to see how the ESS of IS scales as you increase the number of intermediate annealing steps. The authors showed that you can achieve linear scaling with AIS, but what is the relationship for IS (is it exponential)?

---

> > > ### Author Response · Authors · 2025-04-04
> > >
> > > Thank you for acknowledging that our work is of interest to the ICML community. We address your remaining concerns in the following:
> > >
> > > 1. To simplify our workflow, we are not reusing samples from previous annealing steps. When transitioning from $T_i$ to $T_{i+1}$, we train solely on $\mathcal{W}_{i+1}$ as defined in Section 4.2. We will make this more explicit in the manuscript and discuss reusing older samples in the appendix, as it may improve efficiency. We suggest monitoring the ESS of past buffers to guide such reuse.
> > >
> > > 2.+3.:
> > > We want to go into detail regarding your concerns with importance sampling (IS), which we agree is important when scaling our approach.
> > >
> > > The problem with IS is that small local bias, e.g., the possibility of two atoms clashing locally in the proposal, gets amplified due to the dimensionality of the problem. We consider a molecular system with $N$ independent neighborhoods, each with a chance $\eta$ for a clash under the proposal distribution. A sample's importance weight is 1 if no clashes occurred, and 0 otherwise. The ESS is $\text{ESS}=\frac{\left(\sum_{i=1}^M w_i\right)^2}{\sum_{i=1}^M w_i^2}=\sum_{i=1}^M w_i $, so $\mathbb{E}(\text{ESS}) = M \cdot (1-\eta)^N$. Thus, the ESS drops exponentially, which captures why IS does not scale well with dimensionality.
> > >
> > > We now clarify our statement that more expressive architectures can mitigate this. A better architecture with less bias will not remove the scaling problem of IS, unless the proposal exactly matches the target, which is unrealistic. But a better model can reduce $\eta$, thus improving the constants in the exponential scaling law. This allows scaling to larger (though not arbitrarily large) systems.
> > >
> > > To demonstrate how AIS addresses this, we move to a continuous model: The proposal distribution is an $N$-dimensional Gaussian with $\sigma=1.1$ and the target distribution is an $N$-dimensional Gaussian with $\sigma=1.0$. This is a simplified model of a single annealing step.
> > >
> > > We first look at vanilla IS in this scenario. We plot the empirical ESS as a function of dimensionality in the following figure: https://ibb.co/xd6jxL6
> > > As expected, the ESS drops exponentially, which can also be shown analytically [1].
> > >
> > > Now we turn to AIS. Per intermediate distribution, we use a single step of HMC and scale the number of intermediate distributions $T$ linearly with the dimensionality $N$, i.e., $T=c \cdot N$. In our demonstration here, we use $c=5$. We visualize the ESS as a function of dimensionality and compare with vanilla IS: https://ibb.co/bDD53J4
> > >
> > > The ESS of AIS when scaling the number of intermediate distributions linearly can be kept approximately constant. We further refer to the AIS paper [3], where this is shown generally for any factorizable distribution under the assumption of perfect HMC transitions. Again, we acknowledge the simplifications present in this analysis. However, it still captures the essence of why IS fails for higher dimensions and how AIS can help. While for molecular systems, exact factorizability of the distribution is of course not given, AIS can still remove atom clashes, etc., which are often local effects.
> > >
> > > We also point to a recent publication [2], which coincidentally worked on very similar molecular systems as in our work. While they train on data from MD, we cover the data-free case, so the task itself is different. However, they use Sequential Monte Carlo (SMC) for sampling from the flow for evaluation. SMC is a variant of AIS where one resamples during the annealing if the ESS drops too low. As you can see from Tables 2 and 3 in [2], the ESS can be kept very high also for the hexapeptide, though we acknowledge that the involved resampling during the SMC in [2] makes judging and comparing these ESS values somewhat difficult.
> > >
> > > Finally, we evaluated the ESS with AIS for the molecular systems in our work. We report the change from using IS (Table 1 in our paper) to using AIS in the following:
> > > - Dipeptide: 95.6% $\rightarrow$ 98.8%
> > > - Tetrapeptide: 62.5% $\rightarrow$ 84.6%
> > > - Hexapeptide: 14.8% $\rightarrow$ 40.0%
> > >
> > > We note that this is for a fixed number of intermediate distributions (8, the same as used in FAB) and the ESS can thus be further increased.
> > >
> > > Overall, we are confident our method can scale, based on approaches as the one we illustrated above. We want to reiterate that scaling to the hexapeptide in this domain is already a big achievement, as previously only a single method (FAB) succeeded even on alanine dipeptide. We are actively aiming to push the field of variational inference in the domain of molecular sampling forward, which we support by releasing the two additional benchmark datasets. We hope our extended AIS analysis motivates a reevaluation of your score.
> > >
> > > [1] Chatterjee & Diaconis 2017, “The Sample Size Required in Importance Sampling”
> > > [2] Tan et al. 2025, “Scalable Equilibrium Sampling with Sequential Boltzmann Generators”
> > > [3] Neal 1998, “Annealed Importance Sampling”

---

### Official Review · Reviewer_CidM · 2025-03-13

**Overall Recommendation:** 3

**Summary:**

This paper considers the problem of off-policy sampling from the unnormalized density distributions and proposes a novel method (TA-BG) based on a normalizing flow architecture (like FAB) that is less prone to mode collapse. In fact, the authors present a way of training a normalizing flow in this setting with the reverse KLD without mode collapse. They start from training a normalizing flow at high temperatures and then annealing the learned distribution into lower temperatures, which prevent collapsing. Moreover, the paper proposes two novel benchmark problems - similar to alanine dipeptide modeling, but harder.

**Claims And Evidence:**

The paper demonstrates that training a normalizing flows with reverse KLD does not necessarily end with mode collapse when we consider higher temperatures. Moreover, the authors present a novel annealing scheme from higher temperatures to lower. The general statements regarding problems with mode collapsing in previous models are reasonable.

The main problem is with the evidence supporting the claims for TA-BG superiority. Firstly, the experimental setup is limited to three datasets, similar to each other. Moreover, the only baselines are FAB, forward KLD, and reverse KLD, even though, FAB is not the only sampler that was tried in alanine dipeptide setting. Finally, the presented results for NLL and ESS for TA-BG and FAB are similar, without any significant difference. More on improving evaluation you can find in the following sections.

**Essential References Not Discussed:**

**Missing references for standard sampling methods:**

[1] Duane, Simon, Anthony D. Kennedy, Brian J. Pendleton, and Duncan Roweth. "Hybrid monte carlo." Physics letters B 195, no. 2 (1987): 216-222.

[2] Hoffman, Matthew D., and Andrew Gelman. "The No-U-Turn sampler: adaptively setting path lengths in Hamiltonian Monte Carlo." J. Mach. Learn. Res. 15, no. 1 (2014): 1593-1623.

[3] Halton, J. H. Sequential Monte Carlo. In Mathematical Proceedings of the Cambridge Philosophical Society, volume 58, pp. 57–78. Cambridge University Press, 1962

[4] Chopin, N. A sequential particle filter method for static models. Biometrika, 89(3):539–552, 2002.


**And novel neural diffusion-based samplers like:**

[1] Vargas, Francisco, Shreyas Padhy, Denis Blessing, and Nikolas Nüsken. "Transport meets Variational Inference: Controlled Monte Carlo Diffusions." In The Twelfth International Conference on Learning Representations.

[2] Sendera, Marcin, Minsu Kim, Sarthak Mittal, Pablo Lemos, Luca Scimeca, Jarrid Rector-Brooks, Alexandre Adam, Yoshua Bengio, and Nikolay Malkin. "Improved off-policy training of diffusion samplers." Advances in Neural Information Processing Systems 37 (2024): 81016-81045.

[3] Zhang, Qinsheng, and Yongxin Chen. "Path Integral Sampler: A Stochastic Control Approach For Sampling." In International Conference on Learning Representations.

**Experimental Designs Or Analyses:**

I think that the experimental design is very limited and should be significantly improved:

- Firstly, FAB is not the only sampling method successfully used for alanine dipeptide, please see, e.g.,  the following papers: [1], [2], [3], and [4]. At least part of them and other samplers should be considered as baselines also (e.g., DDS, DIS, PIS, GFlowNets, or iDEM).
- The evaluation setup is limited to three energies, which are in some sense similar. I will suggest considering also typical sampling problems, e.g., log-cox, DW-4, and LJ potentials.
- A comparison of training costs for the considered baselines would be beneficial, regarding the presented (not significantly better) results of TA-BG.
- Finally, the presented results in Tab. 1 (especially NLL and ESS) don’t support the claims in a convincing way.


**References:**

[1] Holdijk, Lars, Yuanqi Du, Ferry Hooft, Priyank Jaini, Berend Ensing, and Max Welling. "Stochastic optimal control for collective variable free sampling of molecular transition paths." Advances in Neural Information Processing Systems 36 (2023): 79540-79556.

[2] Seong, Kiyoung, Seonghyun Park, Seonghwan Kim, Woo Youn Kim, and Sungsoo Ahn. "Transition Path Sampling with Improved Off-Policy Training of Diffusion Path Samplers." arXiv preprint arXiv:2405.19961 (2024).

[3] Petersen, Magnus, Gemma Roig, and Roberto Covino. "Dynamicsdiffusion: Generating and rare event sampling of molecular dynamic trajectories using diffusion models." (2023).

[4] Phillips, Dominic, and Flaviu Cipcigan. "MetaGFN: Exploring distant modes with adapted metadynamics for continuous GFlowNets." arXiv preprint arXiv:2408.15905 (2024).

**Methods And Evaluation Criteria:**

The evaluation setting is limited in the sense of number of considered energies, their variability, and the considered baselines. However, the proposed novel benchmarks might be beneficial for the community.

**Other Comments Or Suggestions:**

Overall, I think that the current state of the experimental setting is not enough for the general work in sampling community.

For other comments, please refer to the previous sections.

**Other Strengths And Weaknesses:**

**Strengths:**

[1] Proposing a way of annealing the target distribution to lower temperatures.

[2] Showing that training a NF with reverse KLD at high temperatures is possible without mode collapse.

[3] Introducing two novel benchmarks.



**Weaknesses:**

[1] Missing related references and methods (see above).

[2] Limited experimental setup, not supporting the claims.

[3] Lack of comparison with other than FAB baselines, especially worth to consider are diffusion-based samplers.

**Questions For Authors:**

**Questions:**

[1] Since the proposed method needs an annealing procedure from high to low temperatures, it seems to be computational heavy. Could you present the computational cost of TA-BGs and compare against baselines? I think that considering both training and inference time might be beneficial.

[2] I think that comparing against diffusion-based samplers would be beneficial also from the perspective of mode collapse issue. If I remember correctly, it is not such an issue as for Normalizing Flow-based samplers?

For additional questions, please refer to the previous sections.

**Relation To Broader Scientific Literature:**

This paper is positioned within the sampling community, focused on sampling from molecular energies like alanine dipeptide. However, the proposed method seems to be novel, the paper doesn’t compare (theoretically and empirically) against related baselines.

**Theoretical Claims:**

I haven’t found any obvious flaws in the theoretical properties of TA-BG.

---

> ### Author Rebuttal · Authors · 2025-03-31
>
> Thank you for your detailed and helpful feedback. We address your questions and concerns below:
> * While Table 1 suggests TA-BG mainly improves in terms of the number of target evaluations, it’s important to note that the high-energy metastable regions constitute a small part of the state space. Small NLL differences can thus have significant implications. For example, in the hexapeptide system, TA-BG and FAB show similar NLLs, but Ramachandran plots (Fig. 3) reveal FAB fails to resolve the high-energy metastable region, while TA-BG succeeds. Therefore, quantitative metrics should be complemented with system-specific analyses like Ramachandran plots.
> In summary, TA-BG achieves better NLL across all systems, needs significantly fewer target evaluations, and is the only method to capture the complex interactions in the hexapeptide.
> * FAB is, indeed, the only method successfully applied to learning the Boltzmann distribution of alanine dipeptide without prior knowledge (as also confirmed by Reviewer SzWG in his review). Of the works you cited:
>     * [1] and [2] discuss transition path sampling between points A and B, which is a different task than sampling the full Boltzmann distribution.
>     * [3] relies on data from the target distribution; we cover the data-free case.
>     * [4] depends on a low-dimensional collective variable, typically non-trivial to obtain.
> We will cite these papers and discuss that they are related but ultimately solve a different task.
> * While diffusion samplers have gained popularity, they have not yet been successfully applied to alanine dipeptide. Most diffusion sampling studies use synthetic benchmarks with less correlated transitions between minima than the molecular systems we examine. In our reply to reviewer 1DKq we further discuss why the necessity of using Cartesian coordinates might make applying diffusion models to sample molecular systems harder, but we can mostly speculate here. As reviewer 1DKq points out, diffusion models are "not quite there yet", but we hope to see a successful publication on this topic soon. FAB and our method remain the only approaches that scale to system complexities like alanine dipeptide, which we now significantly extended to the tetrapeptide and hexapeptide.
> * We now include the 2D GMM system (as introduced by FAB). Since diffusion methods have been successfully applied here, this allows a direct comparison. We pre-trained the flow at T=30.0 with reverse KLD, annealed to T=1.0 in 7 steps, and performed one fine-tuning step. Results are shown in this figure (neural splines: https://ibb.co/7dkKCr76) and table (https://ibb.co/bgrDkVyD), using values reported by iDEM [5]. We note that we used two different flow architectures, REAL NVP (as originally used in FAB) and an improved neural splines architecture. We did not have enough time in the rebuttals to repeat the FAB experiments with neural splines, but we will prepare this for the revised manuscript (and also include error bars for our method). As shown, TA-BG outperforms FAB and all diffusion baselines on the 2D GMM system. We will further explore other sampling tasks, such as DW-4, and include them in the appendix.
> We emphasize our focus on molecular systems, which present more challenging and realistic benchmarks than commonly used synthetic systems. Our goal is to motivate the application of sampling methods, including diffusion, to these more difficult and application-relevant domains, supported by our release of the two new benchmark systems.
> * We approximated training time (excluding evaluation) for FAB and TA-BG on the alanine dipeptide system. On our hardware, FAB takes ~18.2h, TA-BG requires ~17.1h. Thus, runtimes are comparable. Furthermore, inference time is identical, since the same model is used for all experiments.
> Force field evaluations in our current setup are relatively inexpensive. With more accurate and computationally costly forces, such as ML-based foundation models or DFT, the evaluation cost becomes dominant. In such cases, the higher sampling efficiency of our method will translate into significantly reduced computational cost.
> * Thank you for pointing out the missing standard sampling and diffusion references; we will revise the manuscript accordingly.
> * We have performed several new ablation studies (see responses to reviewers SzWG and 1DKq).
>
> With this, we hope that we were able to convince you that our method and results are significant achievements and interesting to the sampling community. We hope to see more sampling approaches, including diffusion models, being applied and benchmarked on the complex and more application-relevant molecular systems proposed in our work.
>
> [5] Akhound-Sadegh et al. 2024, “Iterated Denoising Energy Matching for Sampling from Boltzmann Densities”
> [6] Durkan et al. 2019, “Neural Spline Flows”

---

> > ### Comment · Reviewer_CidM · 2025-04-04
> >
> > I would like to thank the authors for their time spent on their rebuttal and answering my concerns. I'm still a little bit concern about the scale of experimental verification. However, I believe that the additional experiments added during the rebuttal should be included in the final version, and significantly improve this submission.
> > I will raise my score accordingly (2 -> 3).

---

> > > ### Author Response · Authors · 2025-04-04
> > >
> > > Thank you for acknowledging our improvements made during the rebuttal and the clarifications we provided regarding your concerns. We will include all additional ablation experiments and the results on the GMM system in the appendix. In particular, the experiments on the GMM system and the comparison with diffusion samplers will make our paper more accessible to readers not familiar with molecular systems. Thank you for this suggestion.
> > >
> > > We appreciate the raised score and thank you for your review of our work, including all suggestions and comments.

---

### Official Review · Reviewer_1DKq · 2025-03-14

**Overall Recommendation:** 4

**Summary:**

The authors propose an iterative training approach for learning normalizing flows to approximate unnormalized densities, such as Boltzmann distributions for physical systems.   The method proceeds to first train a normalizing flow using the mode-seeking KL divergence match a high-temperature target density.   Using samples drawn at the previous temperature reweighted according to the next (lower) temperature, the method proceeds to train on these examples using the mass-covering KL divergence.   The procedure is iterated until reaching the target temperature.


The authors demonstrate that this approach can mitigate mode-seeking behavior of directly training a normalizing flow at the target temperature.   The proposed method is competitive with or outperforms the previous gold standard method (FAB) on tetrapeptide and hexapeptide systems which are more difficult that standard baselines in the literature.

**Claims And Evidence:**

1)
The motivation for training on successively lower temperatures is natural and well presented.
- the proposed method conveniently uses mass-covering / maximum-likelihood KL divergence for training after the first iteration

2)
This benefit of the algorithm is not properly emphasized throughout the paper
- Below eq. 3, the transition to `we focus on the case' leads the reader to question the introduction of Eq. 2-3 and suggests it may not play a key role
- The mass-covering nature of the KL used in training steps after the first likely plays a key role in the lack of mode dropping.
- This change in loss, facilitated by iterative sampling, should be emphasized in "Training by Energy" and the introduction to Sec. 4.2

3)
The presentation of the method could be improved:
- clarify that the same parameters are used to retrain at each step (i.e. $\theta_{T_i}$ used to initialize $\theta_{T_{t+1}}$ and then discarded)
- an inline equation specifying the temperature schedule would be appreciated
- it is not 100% obvious what is the meaning of a 'fine-tuning step' in Lines 254-260 R.   It would be beneficial to highlight the "off-by-one" nature of the proposal samples in the stated algorithms, how the "fine-tuning" corrects this, and that hexapeptide uses e.g. 2x the number of steps



Most additional findings are empirical rather than theoretical.

**Essential References Not Discussed:**

*Not essential*, but I noticed concurrent work also proposing to train a sampler from a higher temperature target distribution [1].  Their approach is not iterative and involves Sequential Monte Carlo resampling to reweight back to the target temperature along a diffusion trajectory.

The authors might also consider advances in normalizing flow architectures demonstrated in [2]

[1] Skreta et. al 2025, "Feynman-Kac Correctors"
[2] Zhai et. al 2024, "Normalizing Flows are Capable Generative Models"

**Experimental Designs Or Analyses:**

see above

**Methods And Evaluation Criteria:**

The method and evaluation measures make sense for the problem at hand.


I might be interested to see how the ESS of the sampled datasets changes over iterations (with and without 'fine-tuning').


The authors appear to use a large number of training steps at the initial mode-seeking KL training run.   This makes sense since later sampling results depend crucially on good initial learning, but could be emphasized.



4)
I am slightly confused by the philosophy behind setting training parameters.   In line 334-336 L, "our approach achieves better results... while requiring $3.08 x 10^8$ target evaluations,"  it is somewhat unclear what is meant by "requires".   How is the number of gradient steps decided?   Eventually, the authors approximately match the number of function evaluations for forward KL (MD steps), FAB (AIS steps), and the proposed method (resampling), but it might be good to emphasize that the proposed training framework allows for many more gradient steps for the same # PE Evals than comparison methods.

**Other Comments Or Suggestions:**

I personally dislike the "forward" and "reverse" KL nomenclature and don't intuitively follow it when reading papers.   This further emphasizes the need for the authors to clearly state the difference in training objectives in (i) their first step and existing methods, versus (ii) training steps at subsequent temperatures.

**Other Strengths And Weaknesses:**

The experimental results are promising, but the presentation can be greatly improved.

**Questions For Authors:**

The FAB method applies just as easily for α=1 (mass-covering KL) as it would for α=2 (perhaps requiring fewer PE Evals for shorter annealing).   Since the proposed method also optimizes this loss, it may be an interesting further benchmark.

5)
Do the authors have any comment on the use of internal coordinates versus Cartesian coordinates?  I see that Midgley et. al 2023a train from maximum-likelihood on replica exchange MD simulation.    It seems that methods in the field are just "not quite there yet"

6)
How sensitive is the method to clipping the highest importance weights?   The authors throw out the "forward ESS" values due to sensitivity to clipping, but this begs the question of its impact on training and/or Ramachandran plots (although for the latter, the authors helpfully provide reweighting-free evaluations)

**Relation To Broader Scientific Literature:**

The paper is well-positioned in relation to the literature.

In Related Work (Lines 342-348 R), the points about diffusion models should be moved to a new paragraph.
"Our results *suggest* (the possibility) that... *might* benefit" would be better wording.

"computing exact likelihoods for these models is prohibitively expensive"
- this claim should be expanded upon in Lines 191-192 L and/or Lines 345-365 R.    (PF ODE, need for divergences, possible approximations)

**Theoretical Claims:**

None given.

---

> ### Author Rebuttal · Authors · 2025-03-31
>
> Thank you for your positive, detailed, and constructive feedback.
> * We address your points following the same numbering as in your review:
>    2. We will more clearly outline the advantages of our method. For example, explicitly stating the mass-covering nature of the forward KLD and why this is helpful in our case is a great suggestion. We will generally work on making our introduction of forward / reverse KLD more intuitive.
>    3. Thank you for these suggestions to improve the presentation of our method. We will work on these points to revise our manuscript. We will include the equation $T_i = T_\text{start} \left( \frac{T_\text{target}}{T_\text{start}} \right)^{\frac{i-1}{K-1}}$ to clarify the “geometric temperature progression”. Further, we will improve the explanation of the annealing procedure, including the fine-tuning and its motivation.
>    4. Choosing hyperparameters for a fair comparison of the different methods is not straightforward, as the final metrics and the total number of target evaluations form a tradeoff. Thus, you are right that saying our method “requires” a certain number of evaluations is imprecise; we will correct this.
> To better make the tradeoff visible, we included FAB hyperparameter variations in Appendix H of our original submission. Furthermore, we now performed ablation studies to make this tradeoff better visible for our method, see our answer to reviewer SzWG, and the fine-tuning ablation study discussed below.
>    5. One of the reasons why using Cartesian coordinates might make the problem harder is that with internal coordinates, we can constrain the angles and bond lengths to reasonable bounds. This is not possible with Cartesian coordinates, where even completely different chemical graphs can be formed. We suspect this is partly why diffusion-based sampling methods have not yet been successful in modeling molecular systems, since handling periodic torsions is less straightforward with diffusion than with flows, and thus Cartesian coordinates are often used.
>    6. Not clipping the importance weights results in numerical outliers visible in the reweighted Ramachandran plots. Therefore, we clipped the weights when determining the KLD of the Ramachandran plots, analogous to FAB.
> We further ran ablation experiments for alanine dipeptide to determine the impact of clipping the importance weights when resampling the buffer datasets during the annealing. The final NLL and ESS stayed within our error bounds, so clipping has little impact here. We will mention this in the revised manuscript.
> * Further responses to your additional suggestions and questions:
>    7. We performed an ablation study to show the impact of fine-tuning. First, we look at the impact of the intermediate fine-tuning for the hexapeptide system. In the following figure, we show the ESS of the training buffer dataset in each iteration for the case with and the case without intermediate fine-tuning (both experiments have a final fine-tuning step): https://ibb.co/XxZJvr0r
> The buffer ESS of the fine-tuning steps that follow each annealing step is much higher, which is reasonable since we here reweight to the same temperature the flow was previously annealed to ($T_{i+1}=T_i$). One can further see that the buffer ESS generally drops down when not using intermediate fine-tuning. We summarize the impact of intermediate fine-tuning on the final metrics for all systems in this table (https://ibb.co/hjXk96b). As you can see, the impact is the largest for the hexapeptide, which is why we included intermediate fine-tuning only in this system for our main experiments.
> We further summarize the impact of the final fine-tuning step for all systems in the following table: https://ibb.co/XfcbxGb3
>    8. Testing other objectives (such as the α=2 divergence) for the annealing is a great idea. However, we believe this goes beyond the main claims we want to cover. We will add a short discussion of other possible objectives to our manuscript.
>    9. Thank you for pointing out the concurrent work [1]. We will mention it in our revised manuscript.
>    10. We will mention more advanced flow architectures as a path for the future [2,3].
>    11. Thank you for your additional suggestions to improve our manuscript. We will include them in the revised version.
>
> In addition to the molecular systems studied in our work, we now additionally benchmarked against the 2D GMM system (see our rebuttal to reviewer CidM), where TA-BG outperforms FAB and all diffusion-based baselines.
>
> Overall, your comments were very helpful in improving our paper. Our approach outperforms the current SOTA (FAB) in this area and demonstrates scalability to more complex systems. We believe that this will be very helpful for the community to advance in this task, and we would appreciate it if you could revise your score based on our revision.
>
> [3] Tan et al. 2025, “Scalable Equilibrium Sampling with Sequential Boltzmann Generators”

---

> > ### Comment · Reviewer_1DKq · 2025-04-03
> >
> > Thanks to the authors for the detailed reply and additional ablation experiments.   I agree that these will notably improve the paper and raise my score to 4.
> >
> > That said, reading other reviews and returning to the paper spurred several other thoughts which the authors may optionally consider.
> >
> > > Testing other objectives (such as the α=2 divergence)... goes beyond the main claims we want to cover.
> >
> > I agree (for the proposed method), but I meant for FAB and α=1.    This is also not essential.
> >
> > > While Table 1 suggests TA-BG mainly improves in terms of the number of target evaluations, it’s important to note that the high-energy metastable regions constitute a small part of the state space. Small NLL differences can thus have significant implications.
> >
> > The above is quoted from the authors' response to Reviewer CidM.   In returning the paper, I also had the question of why NLL performance differences were seemingly so minor, and encourage prominent discussion of this in the final version.
> >
> > > We approximated training time (excluding evaluation) for FAB and TA-BG on the alanine dipeptide system
> >
> > The above is quoted from the authors' response to Reviewer CidM.   I would also be curious to see "total gradient steps" and "wall-clock time" in Table 1.   For gradients, I find myself returning to the appendix, hopping around tables, and doing multiplications to compare.   It is promising that wall-clock time is also favorable or similar compared to FAB, even with retraining.   These are important practical considerations which highlight benefits of the proposed approach.
> >
> > Finally, the schedule ablations reminded me of online adaptive scheduling techniques which use sample-based estimation to approximately construct annealing schedules with constant-rate progress in the Fisher-Rao metric.    See e.g. [1] and discussion of related work within their Sec. 5 (not 100% sure these are applicable).
> > [1] Chopin, Crucinio, Korba, "A connection between Tempering and Entropic Mirror Descent"
> > [2] Syed et. al "Optimized Annealed Sequential Monte Carlo Samplers" (Sec 4.3, 5.1)

---

> > > ### Author Response · Authors · 2025-04-04
> > >
> > > Thank you for raising your score and for the additional suggestions; we will use them in our revised manuscript.
> > >
> > > Since the NLL is not a divergence, the missing scale yields that seemingly small differences can be decisive, which also means that it can be hard to interpret. However, we found that the NLL is still an important indicator, also of mode collapse, and can be complemented with the RAM KLD metric and Ramachandran plots to obtain a more intuitive understanding. We will include an extended discussion on this in our manuscript.
> > >
> > > We agree that a better comparison of the computational cost is called for; we will prepare a table directly comparing compute time, number of flow evaluations, and target evaluations for the revised manuscript.
> > >
> > > Furthermore, thank you for the additional references; the approaches are interesting. We will take a closer look and discuss ways of better annealing schedules in the appendix.
> > >
> > > Again, thank you for your detailed feedback, suggestions, and positive evaluation of our work.

---

### Official Review · Reviewer_SzWG · 2025-03-16

**Overall Recommendation:** 4

**Summary:**

This paper proposes temperature annealed Boltzmann generators. The proposed idea is train a normalizing flow with reverse KL at some high temperature, then train a series of models down to the target temperature using forward KL using generated samples reweighted with importance sampling from a higher temperature model.

It is shown that this method scales up to Alanine Hexapeptide in an internal coordinate system favourably compared to flow annealed bootstrapping and reverse KL training directly at the target temperature.

**Claims And Evidence:**

Yes

**Essential References Not Discussed:**

None

**Experimental Designs Or Analyses:**

Yes

**Methods And Evaluation Criteria:**

Yes

**Other Comments Or Suggestions:**

None

**Other Strengths And Weaknesses:**

Strengths:
* I think this is a very strong work providing a useful method and observations. I also found a number of tricks in the appendix that I personally had not thought about in parametrization. I think this will be a useful work for the community to build off of.
* I found the exposition and figures very clear, although I do think that it could be made slightly clearer to the broader ML community why these are the interesting figures to look at.
* The main results appear strong with adequate baselines.

Weaknesses:
* This work could be made much stronger with additional ablations around hyper parameter choices. I think this paper proves that there exists a setting which scales to at least AL6 using this method, but not a systematic understanding of what it takes to get there or further. In particular, there are a number of hyper parameters which seem to appear without justification and seem like they might be extremely important to the overall performance of the method. Specifically,
  * Initial target temperature (set to 1200K here). It's unclear where this number came from or how it was decided. It seems like this would have a significant effect on results, but it is unclear how much of an effect it has. It would be very reassuring to see that this method also works within a reasonable range of initial target temperatures, or that there is a reasonable method to selecting the initial target temperature.
  * Number of intermediate temperature annealing steps. As far as I can tell all that is mentioned is "For all experiments, we chose 9 temperature annealing steps."
  * Intermediate fine tuning. Very little detail is given on how this is done, what parameters are used, and what effect it has. What is the effect of this empirically?
  * Number of samples drawn for each intermediate distribution.
  * Number of steps trained for / stopping criterion at each temperature.
  * Temperature annealing schedule

**Questions For Authors:**

1. I think the missing baseline is training with reverse KL at the high (1200K) temperature then directly importance sampling down to 300K (and either training a flow or not). I think this would emphasize the need for annealing and the benefit of training many flow models.

2. Would it be possible to include additional ablations I think the most interesting would be on starting temperature and number of temperature scales? I think this would greatly improve the practical applicability of this work.

**Relation To Broader Scientific Literature:**

This paper proposes using reverse KLD at a higher temperature then annealing which is a complementary approaches to previous works. While many works use temperature and many works use normalizing flows in this field I have not seen work that combines these two pieces together in this way.

I agree with the authors assertion that to the best of my knowledge too the only ML approach that has been successfully applied to ALDP is FAB.

**Theoretical Claims:**

No theoretical claims

---

> ### Author Rebuttal · Authors · 2025-03-31
>
> Thank you for your detailed feedback and the many helpful suggestions. We address them in the following:
>
> We performed ablation studies to investigate the impact of hyperparameters, which we will include in our revised manuscript:
> * We ran ablation experiments to determine suitable starting temperatures for each of the systems. At a given temperature, we performed 4 experiments to determine how many of these experiments show mode collapse in the Ramachandran plots when pre-training with the reverse KLD:
> https://ibb.co/5XgHtZYM
> There is a large range of temperatures where reverse KLD training is possible without mode collapse. This ablation was performed with a batch size of 1024 and 250k gradient descent steps. Increasing the temperature beyond the “critical temperature” where the experiments start not collapsing anymore allows us to pre-train cheaper with a smaller batch size and fewer gradient descent steps without mode collapse. A tradeoff exists: When increasing the temperature beyond the “critical temperature”, one can pre-train cheaper, but then the annealing will be more expensive.
> This ablation nicely extends our mode collapse discussion in the paper. The fact that alanine dipeptide can be sampled without mode collapse with the reverse KLD at only slightly increased temperature (375 K) is likely interesting and surprising to the sampling community in itself.
> * We ran experiments to justify the geometric temperature schedule. We compare the geometric schedule with a linearly spaced temperature schedule. In the following figures, we show the ESS of each buffer dataset we train on during the annealing:
> https://ibb.co/TDfMwLbs (alanine dipeptide, one final fine-tuning step)
> https://ibb.co/gbBHb66W (hexapeptide, with intermediate fine-tuning steps)
> The buffer ESS “spikes” visible for alanine dipeptide in the end and for the hexapeptide in-between come from the fine-tuning steps, where $T_{i+1}=T_i$, yielding better overlap. One can see that the geometric temperature schedule yields an approximately constant buffer ESS, whereas the ESS drops down significantly toward the end for the linear schedule (since the temperature steps are too large!).
> * Regarding details on the fine-tuning steps, these are performed with the same parameters as the annealing steps. The only difference is that we do not decrease the temperature, but rather reweight to the same temperature the flow was annealed to in the previous step, so $T_{i+1}=T_i$. We will make this clearer in the revised version of our manuscript.
> We further created an ablation to show the impact of fine-tuning. You can find details on this in our reply to reviewer 1DKq. To summarize, the final fine-tuning step is important for all systems, while the intermediate fine-tuning helps keep the ESS up for the hexapeptide.
> * For alanine dipeptide, we ran ablation experiments to show the impact of the number of temperature steps on the final metrics. We summarize the results in the following table: https://ibb.co/NgPHGG9J
> As you can see, a tradeoff between increasing the number of intermediate temperatures (which yields more target evaluations) and improving the final metrics exists.
> In the following figure, we further show the buffer ESS over the course of training: https://ibb.co/Z6VGsLFj
> One can see that with smaller temperature steps, the buffer ESS during training increases.
> * For alanine dipeptide, we ran ablations to show the impact of the number of samples drawn in each annealing iteration (https://ibb.co/jxwVY4M). Again, a tradeoff is visible between the number of target evaluations and the final NLL / ESS.
>
> Furthermore, thank you for suggesting the additional baseline of reweighting directly from 1200 K to 300 K. You can see what this looks like (using $10^6$ samples) in the following figure: https://ibb.co/7JHtBb2v
> We will include this additional baseline in our main text.
>
> In general, the ablations show that a drop in ESS can be counteracted by using more intermediate temperatures, drawing more samples, or with additional fine-tuning steps. We are thus confident that we can scale our method to even larger and more complex systems, but we believe that already scaling to the hexapeptide is a significant milestone that we want to share with the community.
>
> In addition to the molecular systems studied in our work, we now additionally benchmarked against the 2D GMM system (see our rebuttal to reviewer CidM), where TA-BG outperforms FAB and all diffusion-based baselines.
>
> Again, thank you for your helpful comments, we believe to have addressed all suggestions and concerns of your review. The suggested ablations shed light on interesting dependencies present in our method. We are confident that our results, including the suggested ablations, are interesting for the sampling community and provide a foundation for future work. We hope that you can raise your score accordingly.

---

> > ### Comment · Reviewer_SzWG · 2025-04-03
> >
> > I thank the authors for their hard work during the rebuttal period which has led to significant improvements in understanding around this work.
> >
> > > This ablation nicely extends our mode collapse discussion in the paper. The fact that alanine dipeptide can be sampled without mode collapse with the reverse KLD at only slightly increased temperature (375 K) is likely interesting and surprising to the sampling community in itself.
> >
> > Agreed. This is quite surprising to me, and quite useful.
> >
> > I'm somewhat less confident that this will scale to much larger systems, but I believe the results thus far are already a very useful step.
> >
> > The authors have satisfied my concerns with he current work. For this reason I raise my score 3->4.

---

> > > ### Author Response · Authors · 2025-04-04
> > >
> > > Thank you for raising your score and acknowledging the importance of our work, including the added ablations.
> > >
> > > Furthermore, thank you for your detailed review of our work, which included many good suggestions. They not only improved the paper but also deepened our own understanding of the method.

---

### Decision · Program_Chairs · 2025-05-01

**Decision:**

Accept (poster)

**Comment:**

This paper introduces a new training objective for reverse Kullback-Leibler divergence based-training of Boltzmann Generators based on temperature annealing. The work has strong empirical results and the discussion phase strengthen both the ablation studies and comparisons against baselines.